# Unlocking CRISPR-Cas9 editing for widely diverse *Dictyostelid* species

Mireia Garriga-Canut [1,2], Nikki Cannon [1,2,3], Matt Benton[1], Andrea Zanon [1,2,4], Samuel T Horsfield[5], Jacob Scheurich [6], Kim Remans [6], John Lees [5], Alexandre Paix [1,7 ✉] & Jordi van Gestel [1,2 ✉]

## Abstract

**Dictyostelids are a species-rich clade of cellular slime molds that are widely found in soils and have been studied for over a century. Due to a lack of genome editing methods, most molecular research in Dictyostelids has focused on only a single species, Dictyostelium discoideum, which has severely limited broad-scale comparative analyses. Here, we introduce the first CRISPR-Cas9 editing approach that is cloning-free, selection-free, highly efficient, and effective across Dictyostelid species that diverged millions of years ago. Depending on the CRISPR-Cas9 target site, our editing approach generates knock-out efficiencies of up to 90% and knock-in efficiencies of up to 50% without a selective marker. We show that mutants can be isolated as soon as one day post-transfection, vastly outpacing existing methods for generating knock-outs, fusion proteins, and expression reporters. Leveraging single-cell sorting and fluorescent microscopy, we could readily apply our CRISPR-Cas9 editing approach to phylogenetically distant Dictyostelid species, including those that have never been genome edited before. Our methods therefore open the door to performing broad-scale genetic interrogations across the Dictyostelids.**

**Keywords** Genome Editing; CRISPR-Cas9; *Dictyostelids*; *Dictyostelium discoideum*; Cell Sorting
**Subject Categories** Methods & Resources; Microbiology, Virology & Host Pathogen Interaction

## Introduction

*Dictyostelids*, commonly known as social amoebae or cellular slime molds, are a ubiquitous group of soil amoebae that are known for their biphasic life cycle, where cells alternate between a solitary feeding phase—phagocytizing soil bacteria—and a collective dispersal phase—aggregating into spore-bearing fruiting bodies (Raper, 1984; Bonner, 2009). Although there are over a hundred described *Dictyostelid* species (Schaap et al, 2006; Baldauf et al, 2018; Sheikh et al, 2018), only one of them, *Dictyostelium discoideum* (Raper, 1935), has been broadly adopted as a "go-to" model species in several research fields: in cell biology, *D. discoideum* is, for example, used to study chemotaxis (Van Haastert and Devreotes, 2004; Meena and Kimmel, 2017), pinocytosis (Vines and King, 2019; Kay et al, 2024) and phagocytosis (Cosson and Soldati, 2008; Jauslin et al, 2021); in developmental biology, it is used to study pattern formation (Tomchik and Devreotes, 1981; Gregor et al, 2010), slug migration (Francis, 1964), and morphogenesis (Loomis, 2015); and, in microbiology, it is used to explore microbial cooperation (Medina et al, 2019; Ostrowski, 2019), predation (Tsuchiya et al, 1972; Stewart et al, 2022; Steele et al, 2023), pathogenicity (Steinert and Heuner, 2005; Cardenal-Muñoz et al, 2018) and other ecological interactions (Kuserk, 1980; Kessin et al, 1996; Bonner and Lamont, 2005; Landolt et al, 2006). *D. discoideum* is easy to culture and stock, grows rapidly, and is genetically tractable (Eichinger et al, 2005; Fey et al, 2007, 2013). In contrast to wild isolates, lab derivatives of *D. discoideum* can furthermore be grown axenically (Sussman and Sussman, 1967), which simplifies experimental practices.

Since the generation of the first *D. discoideum* knock-out mutant in 1987 (De Lozanne and Spudich, 1987; Loomis, 1987; Witke et al, 1987), genome editing methods have strongly improved (Paschke et al, 2018, 2019) with, among others, the development of selectable markers for standard homologous recombination (Sutoh, 1993), restriction enzyme-mediated DNA integration (REMI) (Kuspa and Loomis, 1992; Gruenheit et al, 2021), PCR-mediated gene disruptions (Kuwayama et al, 2002), and a synthetic biology toolbox (Kundert et al, 2020). More recently, Muramoto and colleagues pioneered the first CRISPR-Cas9 editing protocol in *D. discoideum* (Sekine et al, 2018; Iriki et al, 2019; Muramoto et al, 2019; Yamashita et al, 2021), marking another major leap in genome editing. Their method is based on the expression of an all-in-one CRISPR plasmid, encoding *Streptococcus pyogenes* Cas9 (SpyCas9) and one or more single guide RNAs (sgRNAs), making genome editing cost-effective, easy to implement, and accessible for library construction (Sekine et al, 2018). Not surprisingly, CRISPR-Cas9 editing is now widely adopted by the field (e.g.,

[1]Developmental Biology Unit, European Molecular Biology Laboratory, Heidelberg 69117, Germany. [2]Molecular Systems Biology Unit, European Molecular Biology Laboratory, Heidelberg 69117, Germany. [3]Collaboration for joint PhD Degree between European Molecular Biology Laboratory, Heidelberg and University of Dundee, School of Life Sciences, D'Arcy Thompson Unit, Carnelley Building, Dundee, UK. [4]Collaboration for joint PhD Degree between European Molecular Biology Laboratory, Heidelberg and Heidelberg University, Faculty of Biosciences, Heidelberg 69117, Germany. [5]European Bioinformatics Institute, European Molecular Biology Laboratory, Wellcome Genome Campus, Hinxton, Cambridge CB10 1SD, UK. [6]Protein Expression and Purification Core Facility, European Molecular Biology Laboratory, Heidelberg 69117, Germany. [7]Department of Algal Development and Evolution, Max Planck Institute for Biology Tübingen, Max-Planck-Ring 5, Tübingen 72076, Germany. ✉E-mail: alexandre.paix@tuebingen.mpg.de; jordi.vangestel@embl.de

Jauslin et al, 2021) and led to the generation of the first pooled genome-wide CRISPR-based knock-out library in *D. discoideum* (Ogasawara et al, 2022), complementing previous efforts using REMI-seq (Gruenheit et al, 2021; Stewart et al, 2022).

Despite the enormous progress in genome editing methods, several limitations remain. First, it remains relatively time-consuming to generate targeted knock-out or knock-in mutants in *D. discoideum*: it can easily take a few weeks from producing cloning vectors to isolating mutant cells. This can limit the scope of mutants that can be compared in any one study, especially when comparing mutants individually, using an arrayed format. Second, most genome editing methods have a limited scope and can only be applied to *D. discoideum*, leaving most other *Dictyostelid* species genetically unexplored. In fact, existing methods almost exclusively focus on a handful of axenic strains in *D. discoideum*, including the CRISPR-Cas9 editing methods. This constrains genetic analyses across *Dictyostelid* species, which contrasts the long-standing tradition of performing broad-scale phenotypic comparisons among species, focusing on, for example, the diversity of multicellular phenotypes (Raper, 1984; Schaap et al, 2006; Schilde et al, 2014; Kuzdzal-Fick et al, 2023). To be effective across species, a genome editing method should ideally be (1) selection-free, to preclude the need for antibiotic markers, since *Dictyostelids* strongly differ in their resistance to antibiotics both across growth conditions and between species (Paschke et al, 2018; Narita et al, 2020; Zhu et al, 2023; Kawabe et al, 2009; Schaap, 2011; Kawabe et al, 2015); (2) effective under non-axenic growth conditions, since most *Dictyostelid* species cannot be grown axenically; and (3) require little to no optimization, such as the optimization of cloning vectors, to easily target a broad range of *Dictyostelid* species.

In this study, we present a new CRISPR-Cas9 editing and isolation approach that is both highly time-efficient and effective across species: our method is cloning-free, selection-free, effective in non-axenic growth conditions, and readily applicable to distant *Dictyostelid* species. Building on CRISPR methods developed in other eukaryotes (Doudna and Charpentier, 2014; Paix et al, 2017a, 2017b), we directly deliver the ribonucleoprotein (RNP) complex to *Dictyostelid* cells together with a homology-directed repair (HDR) template using electroporation. This approach significantly improves knock-out and knock-in efficiencies, being on par with the plasmid-based CRISPR-Cas9 approaches, but without the need for cloning or antibiotic selection. Boosted by single-cell sorting, we can furthermore isolate knock-out or knock-in mutants as soon as one day post-transfection, which strongly accelerates the rate at which mutants can be generated. We first optimized our methods for *D. discoideum* and then showcased their impact by generating fluorescent knock-in mutants in six widely diverse *Dictyostelid* species, three of which have never been genetically modified before. Our CRISPR-Cas9 editing approach therefore significantly expands the genetic toolkit for phylogenetically distant *Dictyostelids*, while simplifying those in *D. discoideum*, promoting systematic cross-species genetic comparisons.

# Results

## RNP complex mediates high knock-out efficiencies in *D. discoideum*

Genome editing with CRISPR-Cas9 is mediated by the SpyCas9 endonuclease that, guided by RNA, causes a double-strand break that triggers a cell's DNA repair machinery. Repair either leads to error-prone non-homologous end-joining (NHEJ) or HDR. HDR requires a repair template and therefore allows for specific gene edits. SpyCas9 forms a complex with CRISPR RNA (crRNA) and trans-activating crRNA (tracrRNA), where specificity results from the complementarity between the 20nt crRNA spacer sequence and DNA target (Jinek et al, 2012). In many studies, crRNA and tracrRNA are replaced by a single chimeric guide RNA (sgRNA). The DNA target must be flanked downstream by a short protospacer adjacent motif (PAM) sequence that is recognized by SpyCas9 ("NGG"). Previous work has shown that CRISPR-Cas9 editing is more efficient when cells are transfected with the RNP complex, as opposed to a CRISPR plasmid, allowing for selection-free genome editing (Kim et al, 2014). With the aim of improving genome editing methods for *D. discoideum* and expanding these methods towards other *Dictyostelid* species, we started by testing genome editing efficiencies with the RNP complex in *D. discoideum* AX2 grown under standard axenic conditions.

To quantify knock-out efficiencies, we first generated *D. discoideum* AX2 *act5::mCherry* using standard homologous recombination with antibiotic selection (following Paschke et al, 2018), which constitutively expresses *mCherry* under the control of the endogenous *act5* promoter. We transfected this strain with an RNP complex targeting the 5'-end of the *mCherry* CDS (see Fig. 1A, "Methods" and Appendix Text S2) and quantified the fraction of cells with an *mCherry* knock-out (non-fluorescent cells) two days post-transfection using flow cytometry. As *D. discoideum* is haploid, the fraction of non-fluorescent cells is a direct readout for the fraction of knock-out mutants. Sanger sequencing of isolated clones confirmed that cells without a fluorescent signal had obtained an indel mutation at the *mCherry* target site. To trigger HDR, we co-transfected cells with single-stranded donor oligos that serve as a template for homologous recombination and cause a knock-out by introducing a frameshift mutation and stop codon (Fig. 1; for details, see "Methods" and Appendix Figs. S1, S2, S3, and S4). We varied the concentration of the donor oligo (0–4.8 μM), length of homology arms (14, 28, 56 nt), and insertion sizes (1, 4, 37 nt) (Fig. 1B–D). In the absence of donor oligo, we obtained knock-out efficiencies of ~5%, but with co-transfection of donor oligo, these efficiencies substantially increased. Knock-out efficiencies improved with higher oligo concentrations, longer homology arms, and larger insertion sizes. The highest knock-out efficiencies of ~80% were obtained for 37 bp insertions using homology arms of either 28 or 56 bp (Fig. 1C). The 37 bp insertion includes an HA-tag sequence, frameshift mutation and stop codon (Appendix Table S6). Twofold lower efficiencies were obtained with 1 bp ($44 \pm 1.9\%$) and 4 bp insertions ($41 \pm 4.6\%$), both of which were designed to include a frameshift mutation and stop codon (Fig. 1D). Using both Sanger sequencing (Appendix Fig. S3) and Nanopore sequencing (Appendix Fig. S5 and Appendix Table S3), we confirmed that HDR-mediated CRISPR-Cas9 editing resulted in a scarless integration of the donor oligo at the expected target site, without any off-target integrations. Control experiments where cells were transfected with SpyCas9 and tracrRNA only (i.e., without crRNA) showed no mutations in *mCherry* (Appendix Fig. S3F). Knock-out efficiencies varied between cell batches and CRISPR-Cas9 target sites: two of the three tested targets in *mCherry* gave high knock-out efficiencies, whereas one gave a fourfold lower efficiency (Appendix Fig. S6). Finally, we also compared different

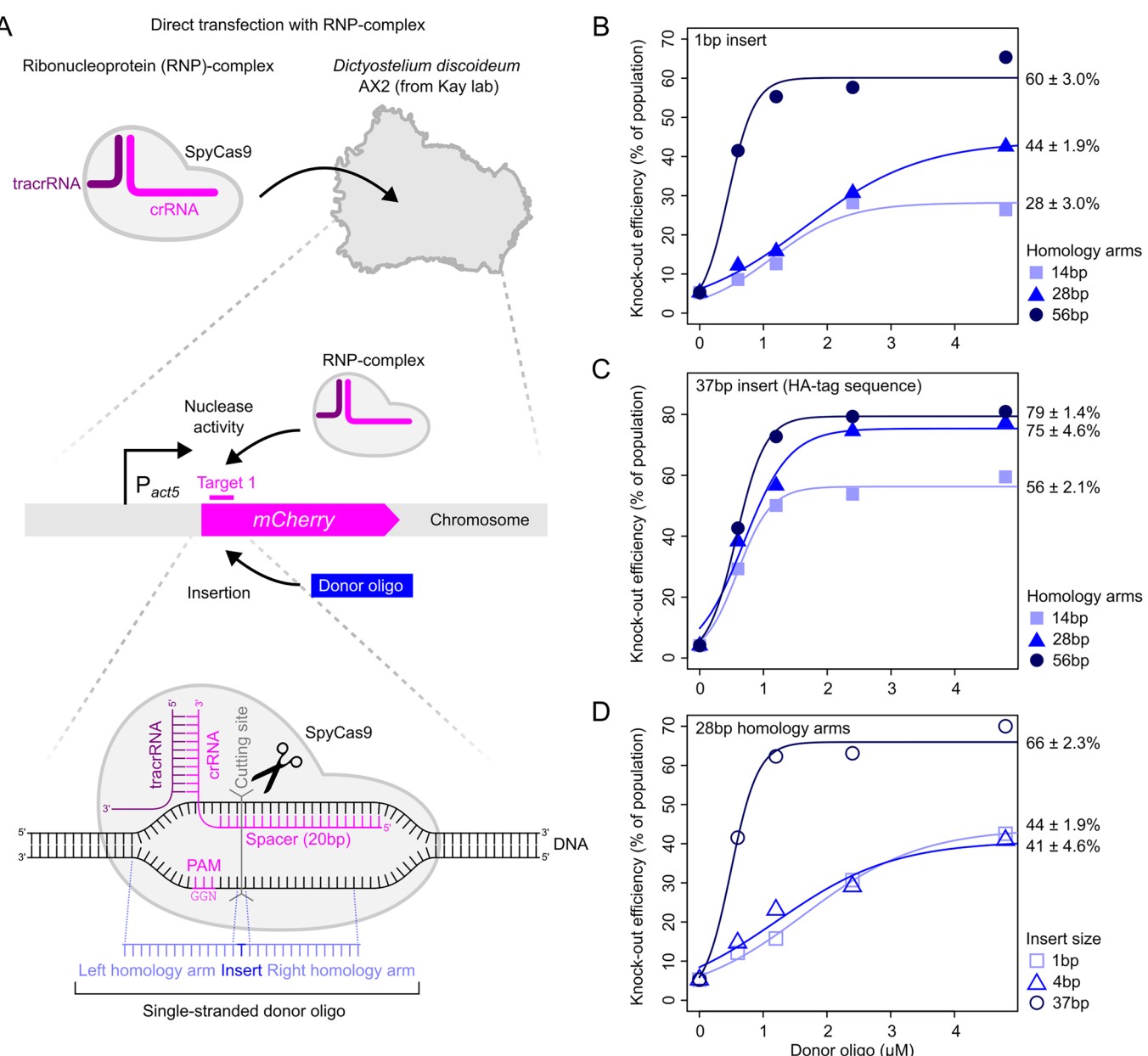

**Figure 1. High knock-out efficiency through donor oligo-mediated CRISPR-Cas9 editing.**

(A) Schematic depiction of CRISPR-Cas9 editing with RNP complex consisting of *Streptococcus pyogenes* Cas9 (SpyCas9), CRISPR RNA (crRNA), and trans-activating crRNA (tracrRNA). Single-stranded donor oligo encodes for an insertion sequence as well as flanking homology arms mediating homology-directed repair (HDR). Knock-outs are produced by targeting amino acid 9 of *mCherry* CDS of *D. discoideum* AX2 *act5::mCherry*. The same target is used for figures hereafter, unless noted otherwise. (B) Knock-out efficiencies for 1 bp insertion (stop codon and frameshift mutation) and 14, 28, and 56 bp homology arms. (C) Knock-out efficiencies for 37 bp insertion (HA-tag sequence, frameshift mutation, and stop codon) with 14, 28, and 56 bp homology arms. (D) Knock-out efficiencies for 1, 4 (stop codon and frameshift mutation) or 37 bp insertion with 28 bp homology arms. Percentages in (B–D) show estimated maximum knock-out efficiencies and standard errors based on sigmoidal fits. As there can be considerable variation between cell batches (Appendix Figs. S1A and S6), the comparisons of knock-out efficiencies within each panel are based on a single batch of cells. The same applies to figure panels with sigmoidal fits hereafter. Knock-outs were generated without pre-annealing the gRNA complex. See Appendix Data S1 for data, and Appendix File S8 for ungated flow cytometry data.

SpyCas9 proteins and concentrations. When comparing wild-type (WT) and high-fidelity variants, and different SpyCas9 purification methods (see Methods), we obtained the best knock-out efficiencies for the ultrapure WT SpyCas9 (see Appendix Fig. S2A and "Methods"). The best-performing commercially available SpyCas9 proteins generated about twofold lower knock-out efficiencies,

possibly due to their lower purity, distinct buffer compositions and/or stock concentrations. For both commercial and ultrapure SpyCas9, knock-out efficiencies linearly correlated with the SpyCas9 concentration (Appendix Fig. S7). Given that knock-out efficiencies peaked for donor oligos with a 37 bp insertion and 28 bp flanking homology arms, we used these oligos at a

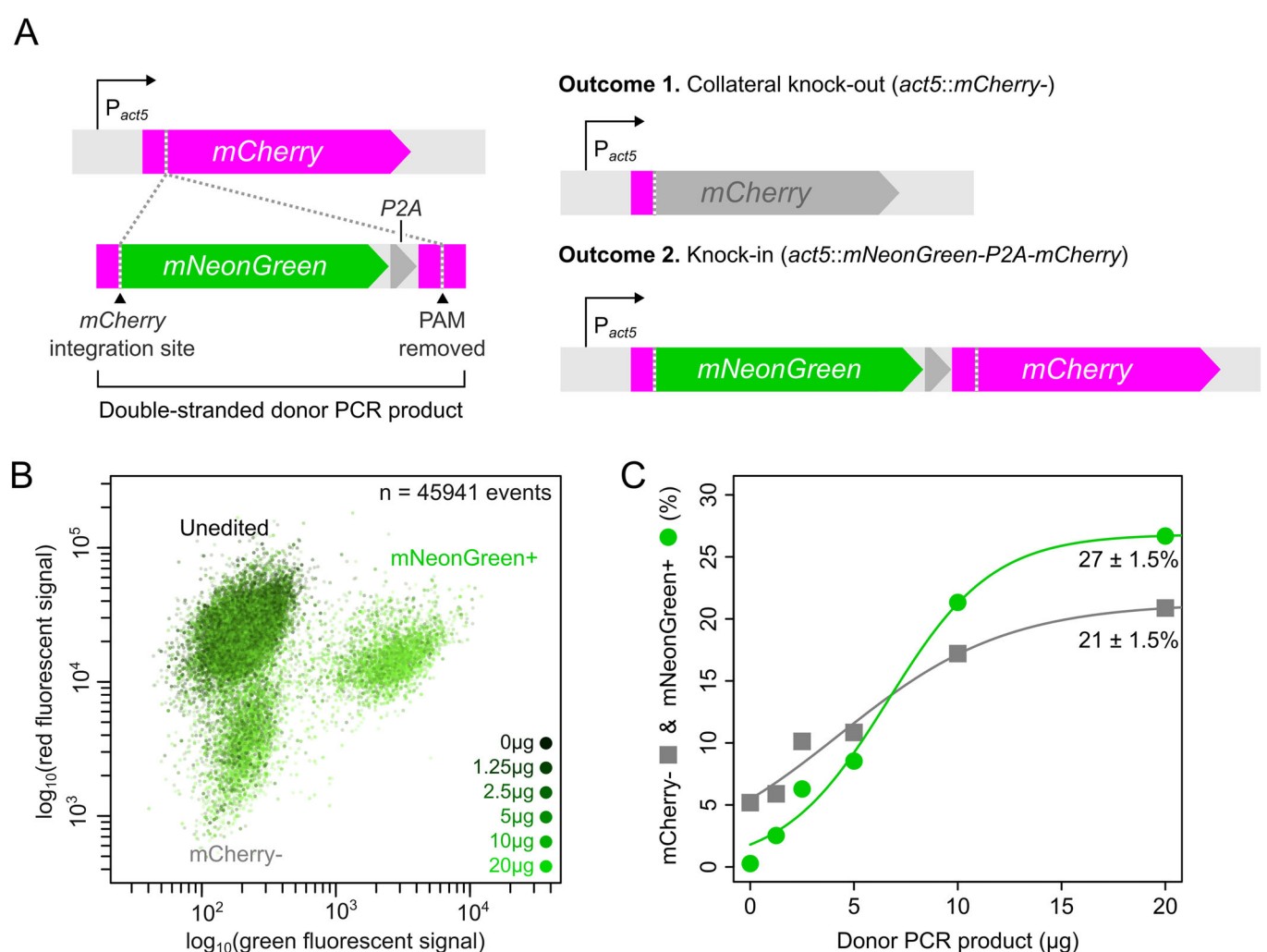

**Figure 2. HDR and NHEJ probabilities for CRISPR-Cas9-mediated knock-in of donor PCR product.**

(A) Schematic of the CRISPR-Cas9 editing approach where *mNeonGreen-P2A* is knocked into the *mCherry* locus, allowing for both green and red fluorescence expression from a single *act5* promoter. Two possible outcomes of CRISPR-Cas9 editing are either collateral knock-out, through, for example, NHEJ, or an *mNeonGreen* knock-in due to HDR. The same target and donor PCR product is used for all knock-in experiments hereafter, unless noted otherwise. (B) Combined flow cytometry results of fluorescent signals of knock-ins produced with 0 to 20 µg (0–36.12 pmol) of donor PCR product (see also Appendix Fig. S8). (C) Knock-in and collateral knock-out efficiencies with 0-20 µg of donor PCR product (see also Appendix Fig. S9). Percentages with standard error show estimated maximum knock-in and knock-out efficiencies based on a sigmoidal fit to data. Knock-ins were generated without pre-annealing the gRNA complex. See Appendix Data S1 for data and Appendix File S8 for ungated flow cytometry data.

concentration of 2.4 µM along with ultrapure WT SpyCas9 for all knock-out experiments described below, unless specified otherwise.

## RNP complex mediates high knock-in efficiencies in *D. discoideum*

Next, we examined whether the RNP complex could also boost knock-in efficiencies. To this end, we co-transfected cells with a donor PCR product encoding *mNeonGreen-P2A* and target-specific homology arms (34/37 bp) for generating an in-frame knock-in mutation targeting the 5'end of *mCherry* ("Methods" and Fig. 2A). P2A induces ribosomal skipping during translation and can thus be used to express multiple proteins from a single promoter (Zhu et al, 2023). With this construct, we quantified both knock-in and collateral knock-out efficiencies: when a double-strand break in

*mCherry* leads to a full integration of the donor PCR product, there is a gain in green fluorescent signal and when it results in NHEJ or partial integration there is a loss of red fluorescent signal. In the absence of a PCR template, we get ~5% knock-out efficiencies (Fig. 2B,C; Appendix Fig. S8), as observed above. With increasing concentrations of PCR product, we get an HDR-mediated knock-in efficiency of up to 27 ± 1.5% and a collateral knock-out efficiency of up to 21 ± 1.5%. In other words, approximately one-third of the cells are unaffected, one-third have a knock-out mutation, and one-third have the desired knock-in mutation (see Appendix Fig. S2B for knock-in efficiencies of different SpyCas9 proteins).

To determine what causes the increased collateral knock-out rates with increasing template concentrations, we isolated and sequenced tens of collateral knock-out mutants. Many of them resulted from NHEJ-mediated repair (Appendix Fig. S9). NHEJ-

mediated knock-outs can increase with template concentrations due to a carrier effect, where donor PCR product improves transfection rates, stabilize the RNP complex and/or stimulate cell repair mechanisms. In support, we also observe higher knock-out rates for non-specific oligos or electroporation enhancer (Appendix Figs. S9 and S10). Besides NHEJ-mediated knock-outs, we also observed partially integrated donor PCR products, which further increase the number of collateral knock-outs (Appendix Fig. S9E,F). Partial integrations could be caused by small sequence similarities between the *mCherry* target site and *mNeonGreen*, supporting HDR, and/or a degraded PCR product.

As high knock-in efficiencies are obtained with short homology arms (~30 bp, see also Paix et al, 2017a), HDR templates can easily be generated by PCRs using primers with overhangs that match the flanking sequence of the CRISPR-Cas9 cutting site, making our protocol effectively cloning-free. Our RNP-based CRISPR-Cas9 editing protocol thus strongly boosts genome editing in *D. discoideum* by being the first protocol that yields high knock-out and knock-in efficiencies without the need for cloning and a selective marker.

## Genome editing effective in both axenic and non-axenic growth conditions

To target other *Dictyostelid* species, which require bacterial food for growth, our genome editing method should also work under non-axenic growth conditions. Traditional editing methods often show strong differences between axenic and non-axenic growth conditions, because bacteria can interfere with the efficacy of antibiotic selection (Paschke et al, 2018). We therefore next compared how axenic and non-axenic growth conditions, both pre- and post-transfection, affect our selection-free CRISPR-Cas9 editing approach.

Relative to axenic growth (66 ± 3.8%; Fig. 3A), we observed high knock-out efficiencies for non-axenic pre-culturing conditions (Fig. 3B): when growing cells on SM5 agar plates with *E. coli* B/r before transfection and recovering them in HL5, we obtained a maximum knock-out efficiency of 91 ± 3.4%. Recovery on SM5 agar plates with *E. coli* B/r produced somewhat lower knock-out efficiencies (74 ± 4.9%; a 1.2-fold reduction), while recovery in KK2 phosphate buffer with different *E. coli* B/r densities ($OD_{600} = 1$ or 2) showed a further twofold reduction in efficiencies (Fig. 3A,B). The negative impact of non-axenic recovery conditions on knock-out efficiencies, especially in phosphate buffer, suggests that cells with and without edit are in different physiological states and therefore show differential recovery. When doing the same experiment with *D. discoideum* NC4 (Fig. 3C), a non-axenic relative of *D. discoideum* AX2 (Sussman and Sussman, 1967; Bloomfield et al, 2015) with a near-identical genome (Bloomfield et al, 2008), we also obtained high knock-out efficiencies (88 ± 8.3%) under non-axenic growth conditions (i.e., SM5-agar plates with *E. coli* B/r before and after transfection).

Non-axenic growth conditions also support high knock-in efficiencies for both *D. discoideum* AX2 (20%) and NC4 (13.5%) (Fig. 3D), although lower than those observed under axenic growth conditions. These somewhat lower knock-in rates could suggest that the donor PCR product is subject to degradation under non-axenic growth, potentially due to nuclease activity. To mitigate this potential nuclease effect in the presence of bacteria, we followed Du

and colleagues (Du and Schaap, 2014) and briefly (16 h) exposed *D. discoideum* NC4 to HL5 before transfection. This additional pre-culturing step indeed improved knock-in efficiencies (25% instead of 6%) and lowered collateral knock-out rates (14% instead of 24%) (Fig. 3E). Thus, in contrast to many existing genome editing methods, our RNP-based CRISPR-Cas9 editing protocol shows high editing efficiencies in axenic and non-axenic growth conditions, especially when recovering transfected cells on SM5 agar plates with *E. coli* B/r.

## Single-cell sorting expedites isolation of genome-edited clones

Besides genome editing, also the process of isolation requires optimization when aiming to edit distinct *Dictyostelid* species. Under non-axenic growth, mutants are typically isolated using plaque assays: transfected *Dictyostelid* cells are spread across a lawn of bacteria, where they form small plaques, originating from single cells, from which clones can be isolated within days (Fey et al, 1995; Chen et al, 2007). These clones are subsequently propagated for both genotyping and phenotyping. Although plaque assays are effective, they are often difficult to standardize across *Dictyostelid* species and pose problems when growth-deficient mutants are easily outgrown by WT cells. We therefore explored the possibility of single-cell sorting (Fey et al, 1995; Chen et al, 2007), where transfected cells go through a single-cell bottleneck while sorting, thereby saving considerable time and shielding them from other cells.

In contrast to previous sorting protocols optimized for liquid medium (Fey et al, 1995; Chen et al, 2007), we developed a protocol to sort cells on a SM5-charcoal-agar medium with a lawn of *E. coli* B/r (Fig. 4A), which allows for both fast and high recovery rates of single cells and has the benefit that clones can be phenotyped directly, minimizing downstream genotyping efforts. The charcoal is included to minimize background fluorescence, making it possible to directly visualize fluorescent knock-in mutants. To assess the efficacy of our sorting protocol, we mixed a population of green- and red-fluorescent *D. discoideum* AX2 cells and sorted them according to fluorescent signal into two 24-well plates containing agar medium (Fig. 4B). Already three days after sorting, 35 out of the 48 sorted wells showed robust growth (Fig. 4C,D). The lack of growth in the other wells likely results from shear stress-mediated cell death during sorting. In only a single well, we observed a mixture of green and red cells, while all other wells contained only green or red cells, as confirmed through both microscopy and flow cytometry (Fig. 4C,D; Appendix Fig. S11). This high sorting accuracy (34/35) shows that cell sorting is an effective method for swiftly isolating individual clones, which could easily be applied to any *Dictyostelid* species.

## Generating knock-outs, expression reporters, and fusion proteins for endogenous genes

Leveraging our isolation method, we next targeted endogenous genes for creating knock-out mutants, expression reporters, and fusion proteins. To demonstrate the benefit of our sorting method, we first generated knock-out mutants in *pkaC*, which are defective in fruiting body development and can therefore easily be identified after sorting (Appendix Fig. S12). We designed crRNAs for three

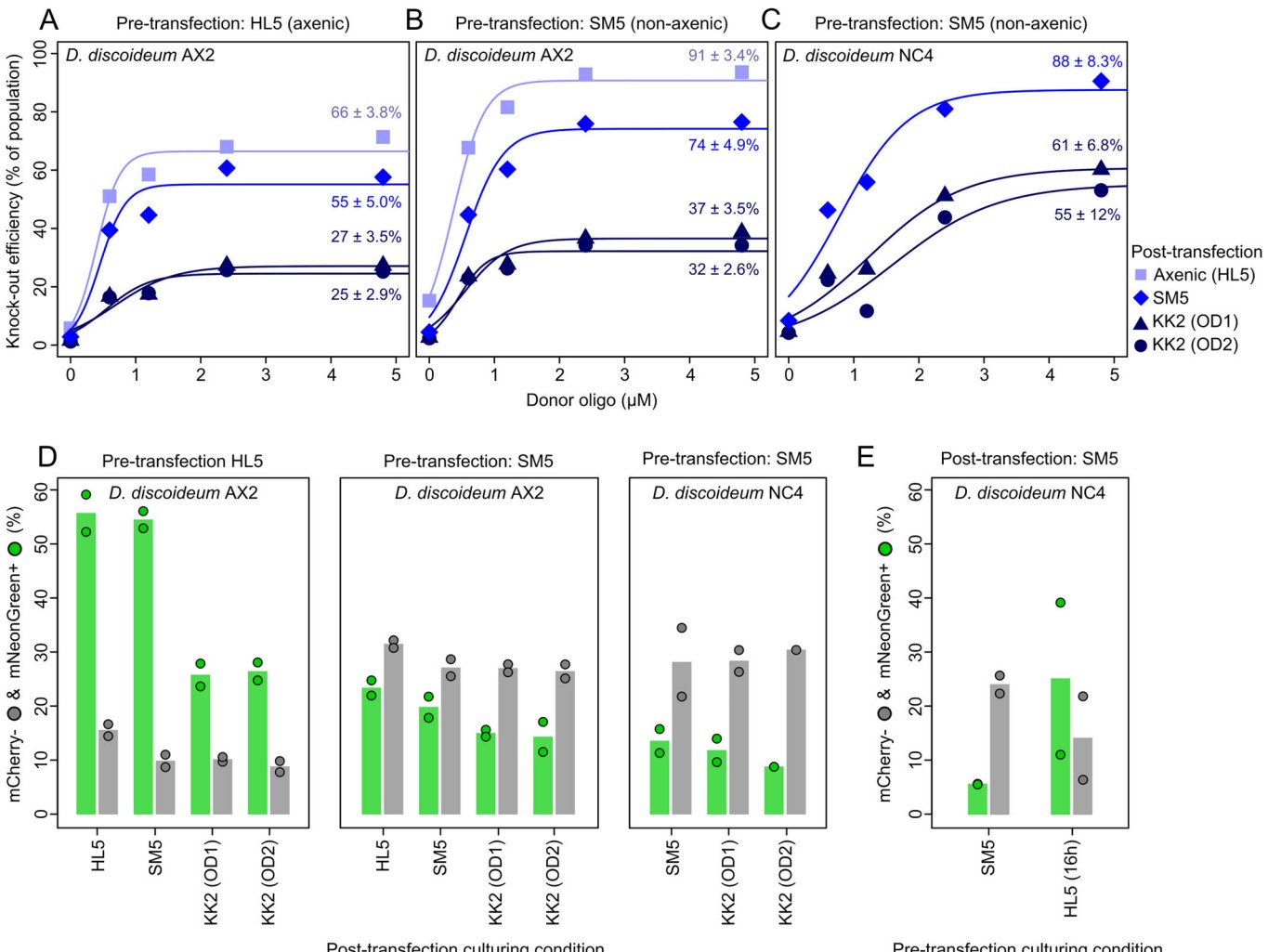

**Figure 3. Knock-out and knock-in efficiencies under axenic and non-axenic growth conditions.**

Knock-out experiment targeting *mCherry* using a donor oligo with 37 bp insertion and 28 bp homology arms. Knock-out efficiencies for (**A**) axenic preculture conditions in *D. discoideum* AX2 *act5::mCherry*, (**B**) non-axenic preculture conditions in *D. discoideum* AX2 *act5::mCherry*, and (**C**) non-axenic preculture conditions in *D. discoideum* NC4 *act5::mCherry*. For recovery, cells were grown either axenically (in HL5) or non-axenically on SM5-agar plates with *E.coli* B/r plates or KK2 phosphate buffer with different densities of *E.coli* B/r ($OD_{600}$ of 1 or 2). (**D**) Knock-in and collateral knock-out efficiencies for *mNeonGreen-P2A* targeting *mCherry* (see Fig. 2) for both axenic and non-axenic growth conditions. (**E**) Effect of knock-in efficiency when shortly pre-culturing (16 h) *D. discoideum* NC4 *act5::mCherry* in HL5 before transfection, instead of using feeding front directly. Knock-out experiments (**A–C**) were performed without pre-annealing the gRNA complex, while knock-in experiments (**D, E**) were done with pre-annealing step. See Appendix Data S1 for data and Appendix File S8 for ungated flow cytometry data.

CRISPR-Cas9 targets within *pkaC*, transfected cells with the RNP complex, and sorted them into 24-well plates (Appendix Fig. S12A). For all targets, we could readily identify wells with arrested fruiting body development (see Fig. 5A as an example well). Of the 12 wells with arrested development, 11 had the expected HDR-mediated knock-out, as confirmed by both PCR and Sanger sequencing (Appendix Fig. S12C and Appendix File S7), and one had a NHEJ-mediated knock-out. Within a week after transfection, we could thus sort, isolate, and genotype the knock-out mutant of interest.

When generating fluorescent knock-in mutants, like expression reporters, genotyping efforts could be simplified even more by directly sorting for fluorescent cells after transfection or by screening for them afterwards using fluorescent microscopy. To test this, we constructed fluorescent expression reporters for *ecmA*

and *act5* by introducing in-frame *mNeonGreen-P2A* knock-ins in their native loci, akin to our knock-in experiment in Fig. 2. EcmA is an extracellular matrix protein expressed by prestalk cells during the slug stage (McRobbie et al, 1988; Jermyn and Williams, 1991). The *ecmA* expression reporter is therefore not expressed during growth, requiring us to sort cells blindly (like for the *pkaC* knock-out above), but can be detected afterwards using fluorescent microscopy. Act5 is a major actin protein that is constitutively expressed during growth (Joseph et al, 2008), which allows for direct sorting of edited cells. All cells were sorted two days post-transfection and imaged a few days later. Within wells, cells form a feeding front, slugs, and fruiting bodies, allowing us to directly examine through fluorescent microscopy whether genome editing was successful. For *ecmA*, 15% of wells with growth showed

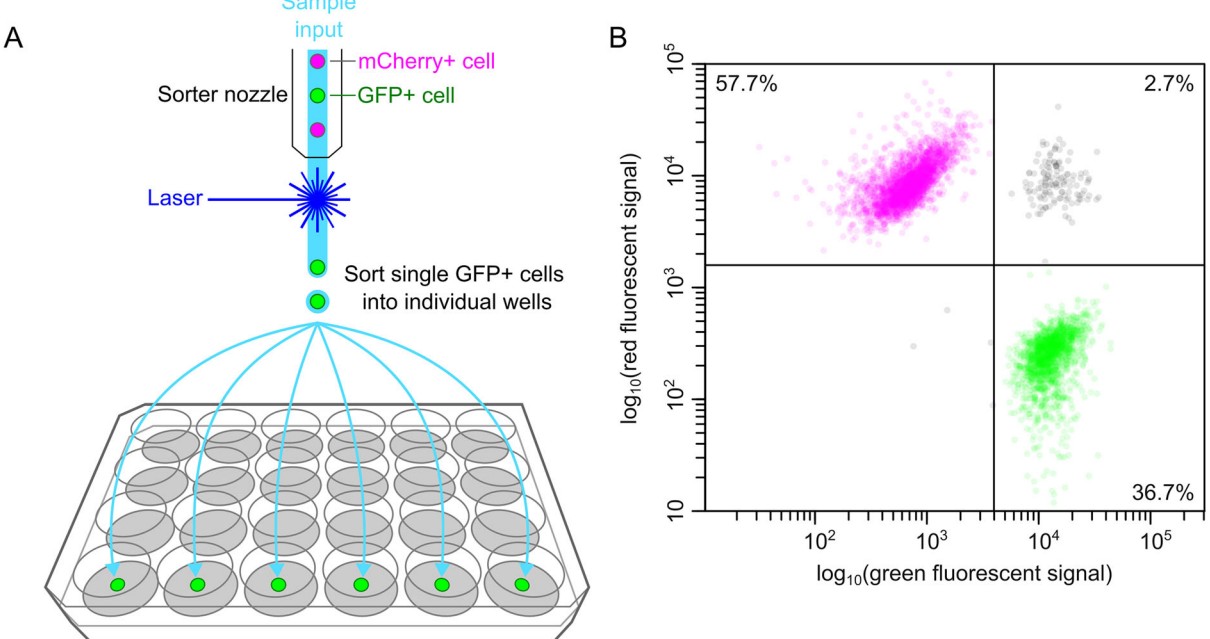

C  Sorted for GFP+ cells

5mm

No growth (3 wells)

GFP+ (21 wells)

D  Sorted for mCherry+ cells

5mm

Mixed (1 well)

No growth (10 wells)

mCherry+ (13 wells)

**Figure 4. Sorting accuracy of red and green fluorescent *D. discoideum* cells across 24-well plates with charcoal agar and *E. coli* B/r.**

(A) Schematic depiction of the sorting procedure, with the sorter nozzle maintaining a constant flow to sort single cells into wells of a 24-well plate based on their fluorescent intensity. Each well contains 1 mL of SM5 charcoal agar with a lawn of *E. coli* B/r. (B) We sorted a mixed population of ~60% mCherry+ cells and ~40% GFP+ cells. From the recorded events, 2.7% showed mixed signal, indicating doublets of green and red cells. (C, D) 24-well plates after 4 days of growth for (C) green and (D) red fluorescent (magenta) sorted cells, respectively. Pie diagrams show sorting accuracy: wells show either no growth (black), GFP+ (green), mCherry+ (magenta) or mixed (gray) cells. For (C), we obtained zero mixed wells and for (D) one mixed well after sorting. See Appendix Fig. S11 for a detailed comparison to flow cytometer data.

expression of our reporter (see example well in Fig. 5B and Appendix Fig. S13A), while for *act5* all wells with growth had green cells (see example well in Fig. 5C and Appendix Fig. S13B). Genomic edits were confirmed by PCR (Appendix Fig. S13C,D) and there were no indications of gene silencing upon perpetual propagation of individual clones. When profiling slugs, we confirmed that, as previously reported, *ecmA* is expressed in the ~20% anterior part of slugs (Loomis, 1987; Jermyn and Williams, 1991), irrespective of their size, while *act5* was uniformly expressed throughout the slug (Fig. 5D,E).

Finally, we also applied our genome editing approach for creating endogenous fusion proteins. For this, we targeted *Act5* and *H2Bv3* and created either N-terminus or C-terminus fusion proteins with *mNeonGreen*. H2Bv3 is one of the main histone proteins involved in chromatin structure and thus localizes to the nucleus (Stevense et al, 2011), while Act5 polymerizes near the leading-edge of moving cells (Ishikawa-Ankerhold and Müller-Taubenberger, 2019). As for the above experiments, we obtained high knock-in efficiencies, especially when targeting the N-terminus (Fig. 6A). Following our expectation, mNeonGreen-H2Bv3 fusion protein localized to the nucleus, while mNeonGreen-Act5 was cytoplasmic (Fig. 6B). When quantifying mNeonGreen-Act5 expression along the radial axis of cells ("Methods"), from their periphery to the center, we observed that—like expected—most Act5 localized near the periphery (Fig. 6C,D).

In summary, our CRISPR-Cas9 editing approach in combination with single-cell sorting makes it possible to target endogenous genes and create knock-outs, expression reporters, and fusion proteins within days: cells can be sorted 1–2 days post-transfection and imaged 3–4 days later. Since our method is cloning-free, new edits can be generated quickly by simply ordering new crRNAs and donor PCR oligos with matching homology arm overhangs (i.e., there is no need to re-design a cloning vector), thereby minimizing preparation time. This strongly contrasts with traditional genome editing methods, where it can take several weeks to construct and examine new knock-in or knock-out mutants (requiring cloning, antibiotic selection, and plaquing).

## Genome editing of multiple genes simultaneously

Since CRISPR-Cas9 editing is targeted and does not rely on selective markers, we hypothesized that it should also be possible to create multiple knock-ins simultaneously. To test this, we mixed the RNP complexes targeting both *act5* and *ecmA*, and performed a single transfection with *mCherry-P2A* and *mNeonGreen-P2A* as repair templates to create a double expression reporter for *act5* and *ecmA*, respectively (Appendix Fig. S14A). From the 22 wells we imaged after sorting for red-fluorescent cells (Appendix Fig. S14B), we detected double knock-in mutants in 13 wells (59% efficiency; Appendix Fig. S15). These mutants showed the characteristic expression of *ecmA*

in the anterior part of slugs, while *act5* was expressed everywhere (Fig. 5F). When targeting *ecmA* only, lower knock-in efficiency was obtained (7/16 wells = 43%, Appendix Fig. S15), suggesting that cells with an *mCherry* knock-in were more likely to also obtain an *mNeonGreen* knock-in. Double knock-ins can also be generated with commercial SpyCas9 protein, although at a somewhat lower efficiency (20%, Appendix Figs. S14 and S15). We did not find any cross-integrations of donor PCR products, despite mixing them within the same transfection (Appendix Fig. S15 and Appendix File S7). When performing a similar experiment for generating a double knock-out using distinct donor oligos (Appendix Fig. S16 and Appendix File S7), cross-integrations were not observed either, confirming the high target-specificity of HDR when providing repair templates with sufficiently distinct homology arms. Given the efficiency of our protocol, the maximum number of fluorescent reporters that can be generated at once is likely to be even higher, especially when these reporters can be sorted using combinatorial gating.

Finally, we also examined whether it would be possible to enrich for knock-out mutations by simultaneously generating fluorescent knock-in mutations for which we can sort. Using *D. discoideum* AX2 *act5::mCherry*, we specifically knocked in *mNeonGreen* at the N-terminus of *H2Bv3* to create an endogenous fusion protein and simultaneously knocked out *mCherry* (without a donor oligo to only allow for NHEJ-mediated knock-out). Like for the double knock-in experiment (Fig. 5F; Appendix Figs. S14 and S15), we observed many cells with both a knock-in and knock-out mutation (Appendix Fig. S17). Double mutants were enriched: cells with a knock-in had a significantly higher probability of also having a knock-out mutation (22% instead of 10%) and, conversely, cells with a knock-out had a significantly higher probability of also having a knock-in mutation (18% instead of 8%) (Fisher's Exact Test; $P < 10^{-10}$, odds ratio = 2.3 with [1.8–2.8] 95%-confidence intervals, $n = 7223$ cells; Appendix Fig. S17). This suggests that edits do not occur randomly but rather that some cells are more permissive to obtain gene edits than others. Combinatorial transfections can therefore be used to enrich for knock-out mutants without immediate phenotype by sorting for parallel knock-ins that do have a phenotype (e.g., protein fusion or expression reporter) (see also Arribere et al, 2014).

As combinatorial transfections require no additional preparation, our CRISPR-Cas9 editing approach strongly simplifies the way multiple genome edits can be generated compared to previous methods, where knock-ins are often produced sequentially by reusing selection markers through Cre recombinase-mediated removal (Faix et al, 2004; Linkner et al, 2012) or where a combinatorial CRISPR-Cas9 plasmid needs to be generated before (Sekine et al, 2018).

## Genome editing effective in many *Dictyostelid* species

With our optimized editing and sorting methods in hand, we next targeted other *Dictyostelid* species. We can only target species for

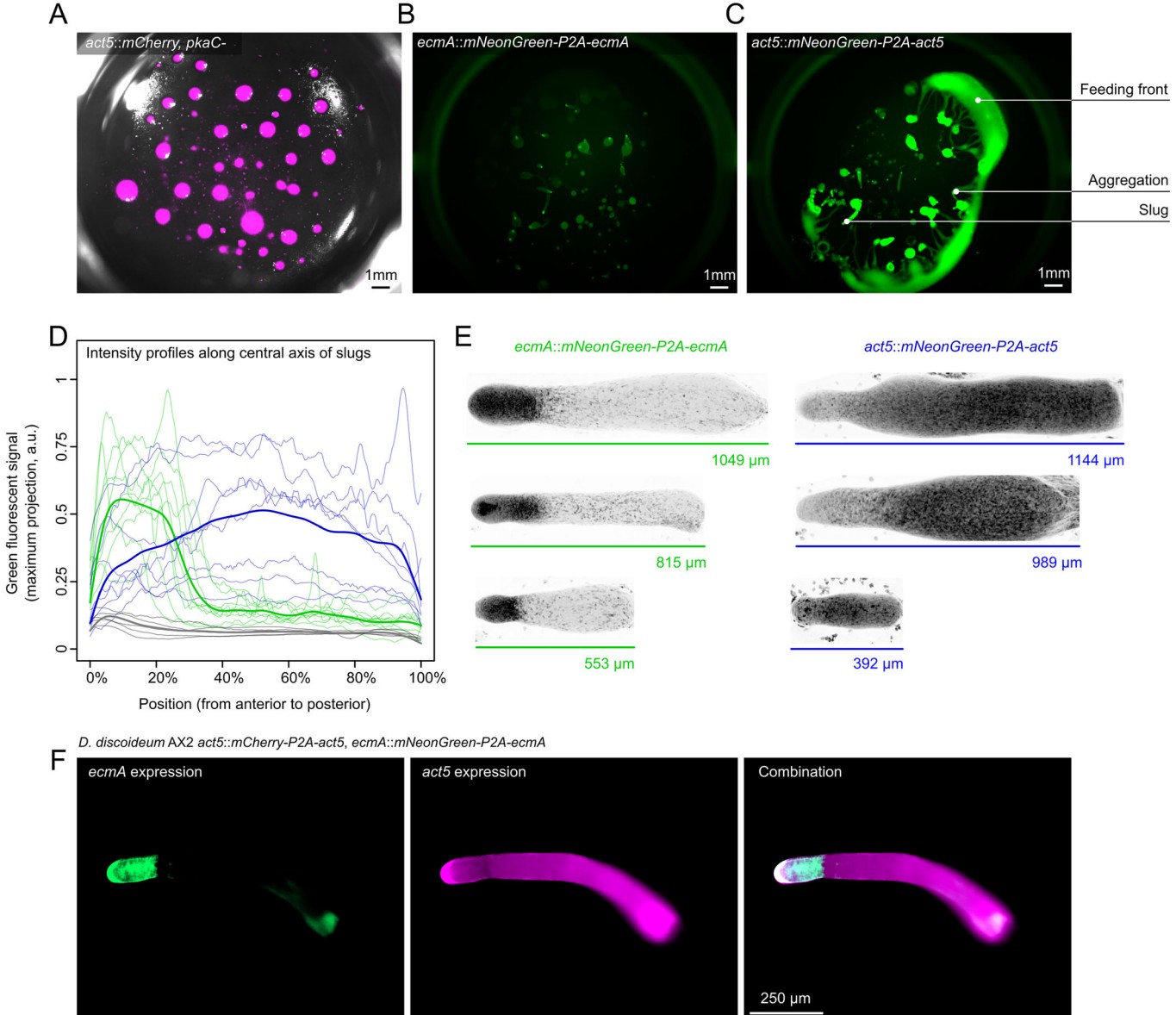

**Figure 5.  CRISPR-Cas9 generated *pkaC* knock-out and *ecmA* and *act5* expression reporters.**

*D. discoideum* AX2 *act5::mCherry* (**A**) or *AX2* (**B–F**) cells were transfected with (**A**) donor oligo (37 insert/28 bp homology arms) targeting the 9th amino acid of *pkaC* (Appendix Fig. S12) or donor PCR products (*mNeonGreen-P2A*) targeting (**B**) the 27th amino acid of *ecmA* (Appendix Fig. S13) and (**C**) the 1st amino acid of *act5* (Appendix Fig. S13). For the expression reporters, genome edits were generated without pre-annealing the gRNA complex. Wells show examples of (**A**) *pkaC* knock-out, (**B**) *ecmA* expression reporter, and (**C**) *act5* expression reporter. (**D**) Expression profiles of slugs, from anterior to posterior, for WT (gray), *ecmA* (green), and *act5* (blue) expression reporters. Transparent lines show maximal projections along central axes of slugs from anterior to posterior, based on a 20 μm rolling window. Solid lines show smooth splines across all maximal projections (*n* = 5–9). (**E**) Example images (with inverse LUT) of slugs for *ecmA* and *act5* expression reporters, confirming that slugs express *ecmA* in the anterior 20% of the slug irrespective of their size (see Files S2 and S3 for image data). Slugs in images are artificially straightened along their anterior–posterior axis to facilitate comparison, but raw image data are provided in File S3. (**F**) Example of double knock-in mutant (*D. discoideum* AX2 *act5::mCherry-P2A-act5, ecmA::mNeonGreen-P2A-ecmA*), where *act5* and *ecmA* knock-ins were generated simultaneously using, respectively, *mCherry-P2A* and *mNeonGreen-P2A* as templates: (left) *ecmA* expression (green), (middle) *act5* expression (magenta), and (right) composite image. See Appendix Figs. S14 and S15 for further details of the double knock-in experiment.

which a genome sequence is available, which we need for identifying unique CRISPR-Cas9 target sites and avoid potential off-target hits. We had access to 15 *Dictyostelid* reference genomes, of which two had the highest assembly level (i.e., chromosome-level assembly; *D. discoideum* and *Dictyostelium firmibasis*). These high-quality genomes are ideal for minimizing the risks of off-target integrations, but lower-

quality genomes (e.g., contig-level assemblies) could in principle be targeted as well, given that HDR relies on short homology arms only.

From all available genomes, we selected eight representative species with *act5* homologs that could be targeted (Fig. 7A; Appendix Fig. S18): *Dictyostelium firmibasis*, *Dictyostelium purpureum*, *Polysphondylium violaceum*, *Tieghemostelium lacteum*,

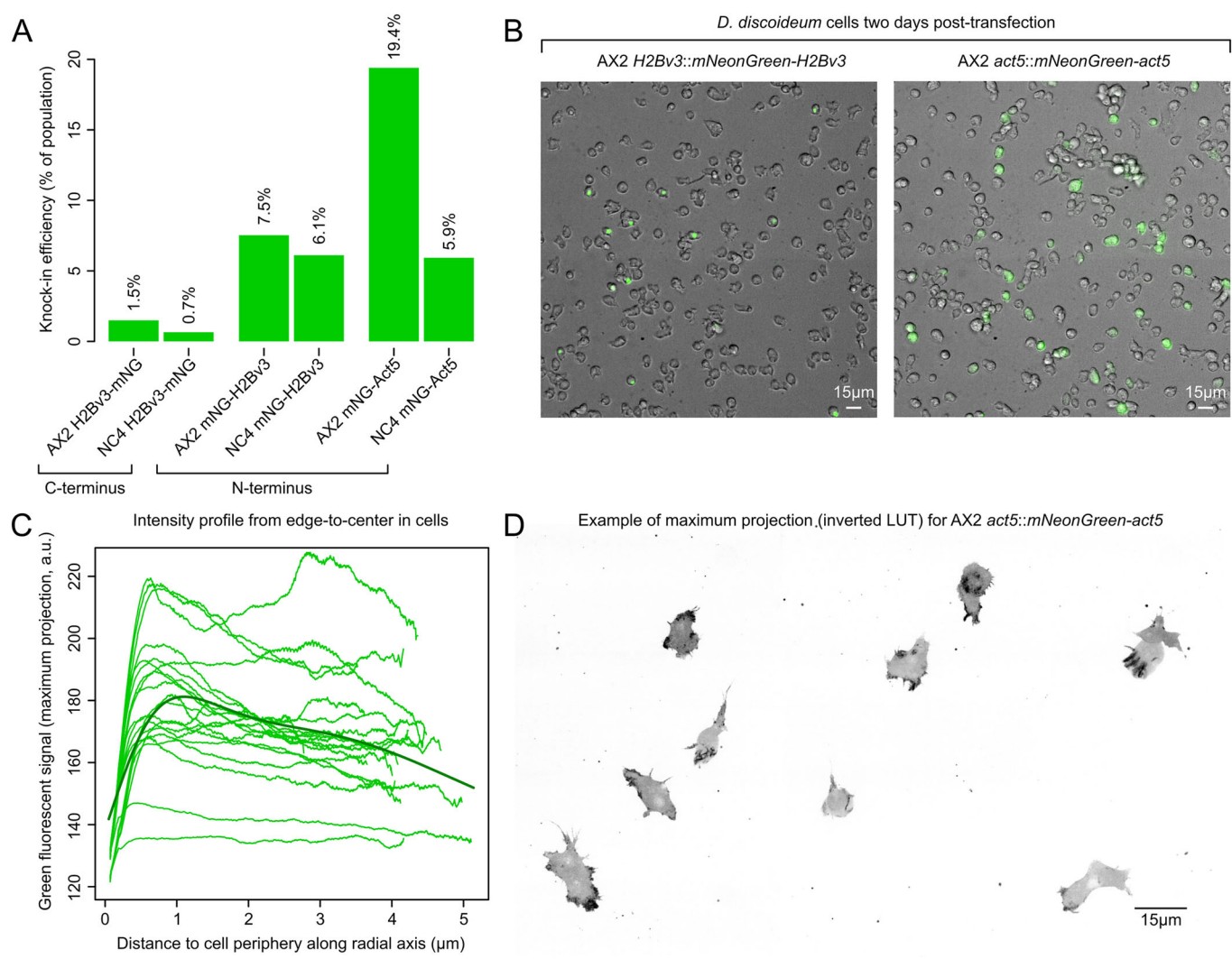

**Figure 6. CRISPR-Cas9 generated H2Bv3 and Act5 protein fusions.**

(**A**) Knock-in efficiencies for *mNeonGreen*, using 10 μg (18.06pmol) donor PCR product, targeting *H2Bv3* and *act5* loci for producing mNeonGreen-H2Bv3, H2Bv3-mNeonGreen, and mNeonGreen-Act5 protein fusions in *D. discoideum* AX2 and NC4. Genome edits were generated without pre-annealing the gRNA complex. (**B**) Cells two days post-transfection, showing nucleus-localized H2Bv3 protein and cytoplasmic-localized Act5 protein. (**C**) Expression profiles of mNG-Act5 along the radial axis of cells from periphery to center, highlighting enriched localization of mNG-Act5 near the periphery. (**D**) Example of mNG-Act5 expression in *D. discoideum* cells (see Appendix File S4 for image data). mNG = mNeonGreen.

*Speleostelium caveatum, Cavendaria fasciculatum, Heterostelium pallidum*, and *Acytostelium subglobosum*. For each species, we directly used our CRISPR-Cas9 editing protocol, without any further optimization, to knock-in *mNeonGreen-P2A* in the N-terminus of the *act5* homolog. Two days post-transfection, we sorted the transfected cells by selecting for green-fluorescent cells. From the eight species, we detected green cells in 6 species, from which we could isolate five species (Fig. 7 and Appendix File S8). For each of those, *mNeonGreen-P2A* was integrated at the expected locus (Appendix File S7). To our knowledge, targeted knock-ins have never been generated for *T. lacteum, D. purpureum,* and *D. firmibasis* before, while for the remaining two (*P. violaceum* and *H. pallidum*) knock-ins were previously generated by integrating antibiotic markers using standard homologous recombination (Fey et al, 1995; Narita et al, 2020).

For all species, knock-in efficiencies were much lower than in *D. discoideum* NC4 (16%), ranging from 0.05% to 11%. Lower efficiencies could result from reduced transfection efficiencies under our electroporation settings, which were optimized for *D. discoideum* (see Appendix Text S2) (Narita et al, 2020), or reduced recovery due to suboptimal growth conditions after transfection (e.g., Fig. 3). For instance, using our electroporation settings, we noticed that most *Speleostelium caveatum* cells did not survive electroporation. Lower efficiencies may also result from lower genome qualities, which can cause errors in designing crRNAs and homology arms and make it difficult to rule out potentially deleterious off-target hits. In support, the *Dictyostelid* species with the best genome assembly, *D. firmibasis*, produced the closest knock-in efficiency (~11%) compared to *D. discoideum*. It could also be that *act5* homologs in other species do not function as a safe

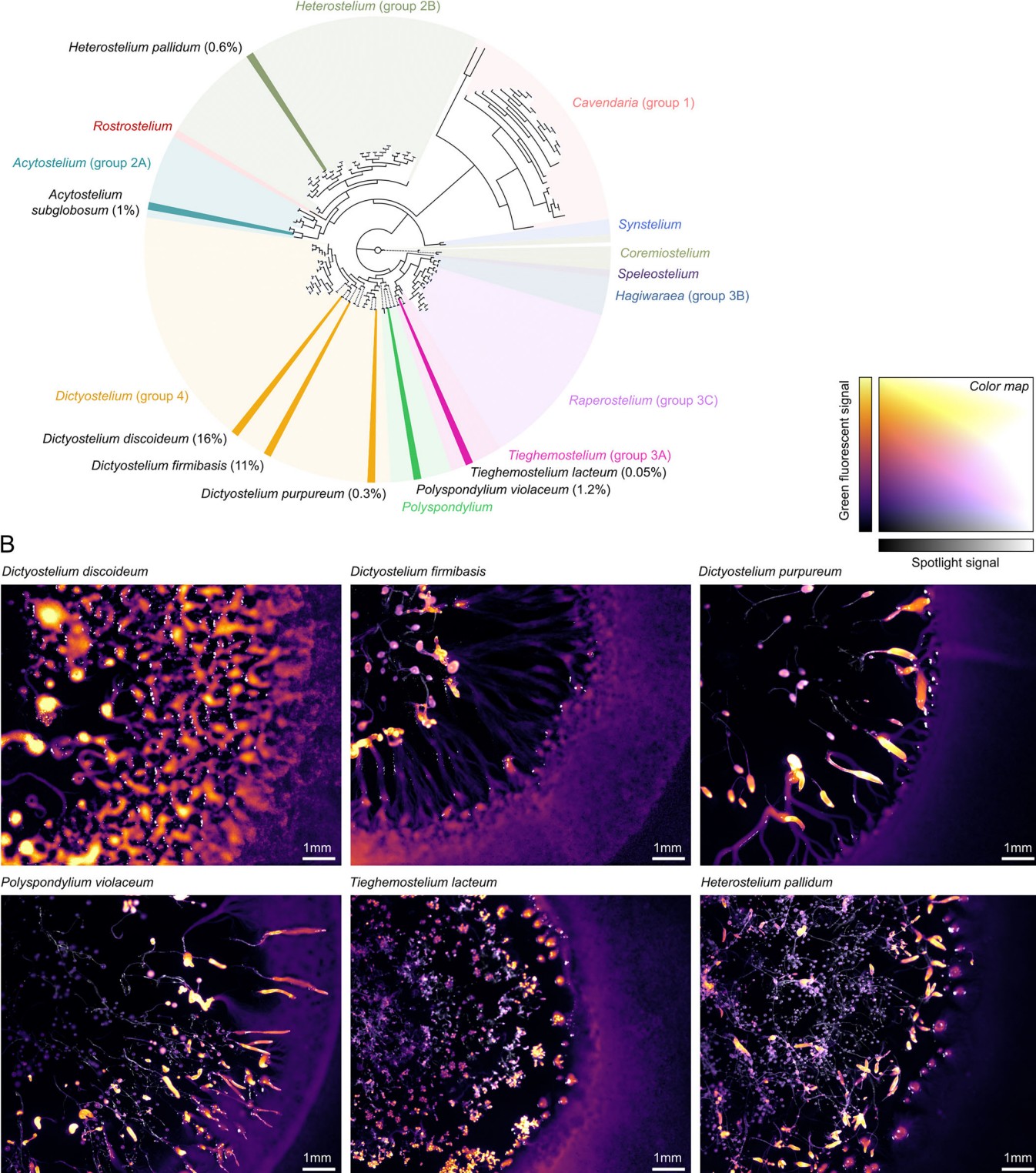

**Figure 7. CRISPR-Cas9 editing of distant *Dictyostelid* species.**

(A) Phylogenetic tree of *Dictyostelids*, based on multiple sequence alignment of 18S rDNA (Sheikh et al, 2018), with highlighted *Dictyostelid* species that were successfully edited (Families are shown in different colors; see Appendix Fig. S18 for detailed phylogeny). Percentages show the fraction of GFP+ cells after knock-in of *mNeonGreen-P2A* in *act5* homolog by CRISPR-Cas9 editing (i.e., knock-in efficiency), without including the pre-annealing step of the gRNA complex. (B) Feeding front for all successfully edited and isolated *Dictyostelid* species: *D. discoideum* NC4, *D. firmibasis*, *D. purpureum*, *P. violaceum*, *T. lacteum*, and *H. pallidum*. *A. subglobosum* was successfully edited, but could not be isolated, likely due to a growth defect. The inferno color scale shows green fluorescence, visualizing the feeding front (falsely colored purple) and aggregation (falsely colored yellow), and the gray scale shows the spotlight. For images, gamma-scaling was applied (see "Methods"). For complete time-lapse data, see Appendix Files S5 and S6. Sanger sequencing results of junction PCRs can be found in Appendix File S7.

locus and lead to growth defects when knocking in *mNeonGreen-P2A*. For instance, in *A. subglobosum*, we observed green fluorescent cells during sorting, but never managed to isolate and propagate those cells afterwards. For the other five edited *Dictyostelid* species we did not observe any noticeable phenotypic defects (Fig. 7B): WT and fluorescent knock-in strains showed similar growth and aggregation dynamics when grown on *E. coli* B/r (see Appendix Files S5 and S6).

In conclusion, despite the lower knock-in efficiencies, our CRISPR-Cas9 editing and isolation methods could readily be applied to distant *Dictyostelid* species, even those that have never been edited before and have low-quality genome assemblies. This strongly expands the genetic toolbox for modifying *Dictyostelid* species. With an increasing number of genomes becoming available, we expect that many more *Dictyostelid* species can be targeted soon as well, making it possible to perform broad-scale genetic interrogations across *Dictyostelid* families.

## Discussion

In this study, we establish a selection-free RNP-based CRISPR-Cas9 editing and isolation approach for *Dictyostelids* that comes with several major benefits (see Appendix Text S2 for a detailed protocol). Compared to existing methods, our editing method strongly expedites the construction of individual mutants in *D. discoideum* for both axenic and non-axenic strains: mutants can be isolated, genotyped, and stocked within a week after transfection. When using donor oligos, our editing approach gives up to 90% knock-out efficiencies (Fig. 3), and with large gene fragments (~1 kb) up to 50% knock-in efficiencies (Fig. 3). For a given gene, efficiencies vary between CRISPR-Cas9 target sites and across cell batches, for the *mCherry* targets this ranged anywhere between 10 and 90% (Fig. 3; Appendix Fig. S6). Editing efficiencies also differ between genes and might in part depend on the chromatin state of cells (Verkuijl and Rots, 2019). Since our approach relies on HDR, knock-outs should be confirmed using Sanger sequencing to rule out potential mis-integrations (see Appendix Figs. S12 and S16). When constructing fluorescent expression reporters or fusion proteins, mutants can be sorted directly based on fluorescent signal, minimizing downstream genotyping efforts. Sorting could also promote the isolation of larger knock-in constructs (>1 kb), which are often associated with lower knock-in efficiencies. By sorting onto agar medium and using fluorescent microscopy, our isolation method also simplifies the isolation of fluorescent expression reporters and protein fusions that are expressed after growth (e.g., during aggregation, slug formation, and fruiting body development). Finally, since our editing method is selection-free, we can

generate several knock-in or knock-out mutations simultaneously, targeting different loci across the genome (e.g., provided that target sites are sufficiently distinct).

Our genome editing method could promote high-content screens in non-axenic *D. discoideum* strains that are difficult to target with existing selection-based protocols. Provided that good CRISPR-Cas9 targets can be identified, proteins from the same pathway or protein family can be interrupted in arrayed knock-out libraries (Bock et al, 2022), complementing pooled and/or non-targeted mutant libraries that exist for axenic strains (Gruenheit et al, 2021; Ogasawara et al, 2022; Stewart et al, 2022). Both donor oligos and crRNAs can be ordered in bulk, using multi-well plates, making it possible to create tens of mutants in parallel. By creating compact and targeted mutant libraries (Bock et al, 2022), one can systematically investigate molecular pathways and protein families. *D. discoideum* could, for example, be used for high-content screens targeting endocytosis (Vines and King, 2019), phagocytosis (Cosson and Soldati, 2008; Jauslin et al, 2021), and chemotaxis (Van Haastert and Devreotes, 2004; Meena and Kimmel, 2017; Jaiswal et al, 2024). *D. discoideum* also shows strong enrichment of important protein families (Eichinger et al, 2005) that can be targeted as well, including polyketide synthases (Zucko et al, 2007), phosphodiesterases (Jaiswal and Kimmel, 2025), ABC transporters (Miranda et al, 2013), actin-binding proteins (Joseph et al, 2008), G-protein coupled receptors (Hall et al, 2022), and tyrosine kinases (Kin et al, 2023). By creating arrayed libraries with tens of knock-out mutants within the same background strain, it becomes possible to systematically interrogate how proteins affect cellular phenotypes, growth, and development.

When targeting a single integration site, our CRISPR-Cas9 editing approach strongly promotes the construction of pooled knock-in libraries with hundreds, if not thousands, of mutants. This makes it possible to generate large-scale gain-of-function, protein-fusion, and expression libraries within a single safe locus (e.g., *act5*). Although plasmid-based expression and gain-of-function libraries in *D. discoideum* already exist (Robinson and Spudich, 2000; Li et al, 2016), the ability to generate knock-in libraries has been limited by low editing efficiencies. With our knock-in efficiencies of up to 50%, our CRISPR-Cas9 editing approach makes it possible to generate ~$10^6$ unique knock-in mutants from a single transfection only, outperforming previous selection-based genome editing protocols. When generating barcoded mutants, large-scale gain-of-function screens can be performed across a wide range of culturing conditions with minimal expenses. Similar gain-of-function libraries have been generated for other bacterial (Urtecho et al, 2023; Huang et al, 2024) and eukaryotic model species already (Jones et al, 2008; Fuqua et al, 2020). Knock-in libraries can also be used for deep

mutational scanning studies to investigate the function of a specific protein. One could, for example, systematically introduce amino acid substitutions along a protein or within a specific protein domain (Papkou et al, 2019; Jones et al, 2020). Given the rich biology of *D. discoideum* and ease of high-throughput culturing, deep mutational scanning methods can be a powerful tool for interrogation key proteins underlying multicellular developmental (e.g., adhesion proteins) and cell biology (e.g., phagocytic receptors).

Finally, without modifications, we showed that our CRISPR-Cas9 editing and isolation methods can be applied to other *Dictyostelid* species, including species that diverged hundreds of millions of years ago and were never genetically modified before. As long as a sufficiently high-quality genome is available and transfection is possible, genome editing should in principle be possible as well. When performing our experiments, only two *Dictyostelid* species with chromosome-level assemblies were available, but our method was also effective for species with lower-quality genome assemblies. With improving long-read sequencing and the rapid expansion of available genome sequences (Holland et al, 2025), we expect that our CRISPR-Cas9 editing approach can soon be applied to tens of *Dictyostelid* species. For some species, transfection conditions might need optimization, including buffer composition and pre-and post-transfection growth conditions, but for most species we expect that our methods would readily work. Our CRISPR-Cas9 editing approach therefore contributes to the increasing number of protists that are accessible through genetics (Faktorová et al, 2020; Nomura et al, 2020; Akella et al, 2021; Combredet and Brunet, 2024). By editing other *Dictyostelid* species, while building on decades of genetic research in *D. discoideum* (Loomis, 2015) and comparative research across *Dictyostelids* (Raper, 1984; Schaap et al, 2006), we can explore how cellular and developmental processes diverged over hundreds of millions of years—making the *Dictyostelids* one of the few phylogenetic groups at which genotypic and phenotypic interrogations can be performed at scale.

# Methods

### Reagents and tools table

| Reagent/resource | Reference or source | Identifier or catalog number |
|---|---|---|
| **Experimental models** | | |
| *Escherichia coli* B/r | dictyBase | DBS0305924 |
| *Dictyostelium discoideum* AX2 (Kay lab) | dictyBase | DBS0235521 |
| *Dictyostelium discoideum* AX2 *act15::gfp* | dictyBase | DBS0235536 |
| *Dictyostelium discoideum* AX2 *act5::mCherry, Hyg* | This study | MG38 |
| *Dictyostelium discoideum* NC4 | dictyBase | DBS0304666 |
| *Dictyostelium discoideum* NC4 *act5::mCherry, Hyg* | Paschke et al, 2018 | HM1912 |
| *Dictyostelium firmibasis* TNS-C-14 | dictyBase | DBS0235812 |
| *Dictyostelium purpureum* WS321 | dictyBase | DBS0235881 |

| Reagent/resource | Reference or source | Identifier or catalog number |
|---|---|---|
| *Polysphondylium violaceum* P6, S209 | dictyBase | DBS0236814 |
| *Tieghemostelium lacteum* S561 | dictyBase | DBS0235831 |
| *Acytostelium subglobosum* LB1 | dictyBase | DBS0235452 |
| *Heterostelium pallidum* PN500 | dictyBase | DBS0302501 |
| *Cavendaria fasciculatum* Swok OW9A | dictyBase | DBS0235811 |
| *Speleostelium caveatum* | dictyBase | DBS0235736 |
| **Recombinant DNA** | | |
| pMG005 | This study. Deposited to Addgene | 237215 |
| pMG008 | This study. Deposited to Addgene | 237216 |
| pHO4d-Cas9 | Addgene | 67881 |
| pET-HiFi SpCas9-NLS-6xHis | Addgene | 207376 |
| pET-FLAG-spCas9-HF1 | Addgene | 126770 |
| **Antibodies** | | |
| **Oligonucleotides and other sequence-based reagents** | | |
| Donor oligos | This study | Table S6 |
| PCR primers | This study | Table S6 |
| **Chemicals, enzymes, and other reagents** | | |
| Agarose | Sigma-Aldrich | A9539-500G |
| Acetic acid (glacial) | Sigma-Aldrich | 1000631000 |
| Activated charcoal | Sigma-Aldrich | C9157-500G |
| Bacto Dehydrated Agar | BD Biosciences | 214010 |
| Calcium Chloride (CaCl$_2$) | Sigma-Aldrich | C3881-500 |
| DRAQ7 | ThermoFisher | D15106 |
| Ethanol | Sigma-Aldrich | 100.983 |
| Folic acid | Sigma-Aldrich | F8758-25G |
| Glycerol | Sigma-Aldrich | 1040572511 |
| HEPES, ultrapure | Biomol | 5288.1 |
| HL5 Medium including Glucose | Formedium | HLG0102 |
| IGEPAL CA-630 | Sigma-Aldrich | I8896-50M |
| Potassium phosphate dibasic (K$_2$HPO$_4$) | Sigma-Aldrich | P8281-500G |
| Potassium chloride (KCl) | Sigma-Aldrich | 60130-1KG |
| Potassium phosphate monobasic (KH$_2$PO4) | Sigma-Aldrich | P0662-500G |
| Magnesium chloride hexahydrate (MgCl$_2$·6H2O) | Sigma-Aldrich | M9272-1kg |
| Magnesium sulfate (MgSO$_4$) | Sigma-Aldrich | M2643-500G |
| Sodium chloride (NaCl) | Sigma-Aldrich | 71380-1KG-M |
| Sodium phosphate monobasic (NaH$_2$PO$_4$) | Sigma-Aldrich | S3139-500G |
| Sodium bicarbonate (NaHCO$_3$) | Sigma-Aldrich | S5761-1KG |
| Penicillin G potassium salt (Benzylpenicillin potassium salt) | Sigma-Aldrich | P7794-10MU |
| SM Broth/5 | Formedium | SMB50102 |

| Reagent/resource | Reference or source | Identifier or catalog number |
|---|---|---|
| Streptomycin sulfate | Sigma-Aldrich | S9137-100G |
| Titriplex III (EDTA disodium salt) | Sigma-Aldrich | 1084180100 |
| Trizma base | Sigma-Aldrich | T1503-1KG |
| Vitamin B12 (Cyanocobalamin) | Sigma-Aldrich | V6629-250MG |
| Nuclease-free water (H$_2$O), for molecular biology (DEPC-free) | Sigma-Aldrich | W4502-10X50ML |
| Bacto Tryptone | ThermoFisher | 211705 |
| Bacto Yeast Extract | ThermoFisher | 212750 |
| Alt-R CRISPR-Cas9 tracrRNA, 20 nmol | IDT | 1072533 |
| Alt-R CRISPR-Cas9 crRNA | IDT | Custom |
| Alt-R S.p. Cas9 Nuclease V3, 100 µg | IDT | 1081058 |
| Alt-R Cas9 Electroporation Enhancer, 10 nmol | IDT | 1075915 |
| Alt-R HDR Enhancer V2, 30 µL | IDT | 10007910 |
| pHO4d-Cas9 (for SpyCas9 purification), see also Ref. (4) | Addgene | 67881 |
| TrueCut Cas9 Protein v2 | ThermoFisher | A36498 |
| QIAquick PCR Purification Kit | Qiagen | 28106 |
| Proteinase K | NEB | P8107S |
| Q5 Hot Start High-Fidelity 2X Master Mix | NEB | M0494L |
| Taq 2X Master Mix - 500 reactions | NEB | M0270L |
| **Software** | | |
| Snapgene | Dotmatics | https://www.snapgene.com |
| Cas designer | RGEN tools | http://www.rgenome.net/cas-designer/ |
| **Other** | | |
| BD FACSAria Fusion Flow Cytometer | BD Biosciences | |
| Gene Pulser Xcell Microbial System | Bio-Rad Laboratories | 1652662 |
| BD FACSymphony A3 Analyzer | BD Biosciences | |
| Axio Zoom.V16 Stereo Zoom Microscope | Zeiss | |
| Countess 3 starter package 1 | ThermoFisher Scientific | A49865 |
| NanoDrop 8000 Spectrophotometer | ThermoFisher Scientific | ND-8000-GL |
| BioPhotometer plus | Eppendorf AG | |
| C1000 Touch Thermal Cycler | Bio-Rad Laboratories | 1851197 |
| Disposable loops 10 µL | ThermoFisher Scientific | 10048750 |
| Disposable loops 1 µL | ThermoFisher Scientific | 10344741 |
| Spreaders, T-shaped | VWR | 612-2653 |
| Nunc multidish 24-wells | ThermoFisher Scientific | 10604903 |
| Nunc multidish 6-wells | ThermoFisher Scientific | 10119831 |
| Petri dishes, Steriplan, Duran 100×20 | Duran | 391-2840 |
| Gene Pulser/MicroPulser electroporation cuvettes, 0.1 cm gap | Bio-Rad Laboratories | 1652089 |
| Countess cell counting chamber slides | ThermoFisher Scientific | C10312 |
| Falcon 5 mL round-bottom polystyrene test tube, without cap | Falcon | 352008 |
| Falcon round-bottom polystyrene test tubes with cell strainer snap cap, 5 mL | Falcon | 352235 |

## Reagents and strains

A complete list of strains, as well as Dicty Stock Center ID (if applicable), is available in Appendix Tables S1, S2, S3, and S4. Selected strains will be deposited to Dicty Stock Center and ATCC. For knock-out and knock-in mutants, we also list the crRNAs, donor oligos, donor PCR template, and primers (for donor and confirmation PCRs) in Appendix Tables S2 and S4. crRNA and primer sequences are given in Appendix Tables S5 and S6. The pMG005 vector (ordered from Geneart) with a codon-optimized *mNeonGreen* and *P2A* peptide sequence (Zhu et al, 2023) is provided in Appendix Text S1 and Appendix File S1. pMG005 was used as a template to create *mNeonGreen-P2A* donor PCR products. pMG008 was used as a template for the *mCherry-P2A* knock-ins and is provided in Appendix Text S1 and Appendix File S1. Plasmids are promoter-less for *mNeonGreen* and *mCherry* to avoid expression of fluorescent proteins from the plasmid directly. Only after donor PCR products are integrated into the chromosomes can *mNeonGreen* and *mCherry* be expressed. pMG005 and pMG008 are deposited to Addgene (IDs 237215 and 237216, respectively). Lists of all reagents, buffers, and other materials are provided as part of the detailed CRISPR-Cas9 editing protocol in Appendix Text S2 as well as in the Reagents and tools table above.

## Strains and culturing conditions

For each experiment, *Dictyostelid* cells were freshly inoculated on SM5-agar plates with *E. coli* B/r from stocks and, for axenic strains, subsequently propagated in HL5 + FAB without bacteria before transfection. SM5-agar plates with a lawn of *E. coli* B/r were prepared as described in Appendix Text S2, Step 3.1. *Dictyostelid* species were directly spotted on two opposite sides of the lawn from glycerol stocks (Appendix Table S1) and grown at 22 °C for 3–5 days, until large feeding fronts were visible. For axenic strains, the feeding fronts of 4 to 6 plates were harvested in HL5 and grown for 2–3 days in 50 mL of HL5 + FAB at a starting concentration of $10^5$ cell/mL at 22 °C (180 rpm) until they reached the exponential phase (0.8–1.2 × $10^6$ cells/mL), as described in Appendix Text S2, Step 3.2. Typically, cells reached exponential growth after 2 days.

## Expression and purification of SpyCas9

Different variants of recombinant SpyCas9 were purified by the Protein Expression and Purification Core (PeP-core) Facility at EMBL using either a quick one-day protocol for regular purity protein extract or an extensive three-day protocol for an ultrapure protein extract. In order to prevent endotoxin contamination in both the quick and long protocol, Åkta chromatography stations were incubated with 1 M NaOH for 4 h before use, all chromatography columns were rigorously cleaned with 0.5-1 M NaOH, and all protein purification buffers were prepared using endotoxin-free reagents.

For the quick protocol, pHO4d-Cas9, pET-HiFi SpCas9-NLS-6xHis and pET-FLAG-spCas9-HF1 (Addgene, Catalog # 67881, #207376 and # 126770 respectively, and Appendix File S1) encoding WT SpyCas9, HiFi SpyCas9 and HF1 SpyCas9, respectively, were freshly transformed into *Escherichia coli* Rosetta2 (DE3) cells. Precultures were grown overnight at 37 °C in LB medium supplemented with 100 µg/mL carbenicillin and 34 µg/mL chloramphenicol and used to inoculate the large-scale expression cultures. 10 mL preculture was added to 1 L of TB-FB supplemented with 2 mM MgSO4, 0.05% glucose, 1.5% lactose, 100 µg/mL carbenicillin, and 34 µg/mL chloramphenicol. Cultures were grown at 37 °C until $OD_{600}$ reached ~0.6, after which the temperature was reduced to 18 °C. After overnight expression at 18 °C, the cultures were harvested by centrifugation (30 min, 5000 × *g*, 4 °C) and pellets were flash-frozen in liquid nitrogen and stored at −80 °C until the start of the protein purification step.

The cell pellet was resuspended in cold lysis buffer (20 mM Tris-HCl pH 8.0, 750 mM NaCl, 20 mM imidazole, 10% glycerol and EDTA-free cOmplete protease inhibitors, Roche). Cells were lysed by 5 passages through a microfluidizer, followed by centrifugation (30 min, 140,000 × *g*, 4 °C). The cleared lysate was loaded onto a 1 mL Protino Ni-NTA column (Macherey-Nagel) pre-equilibrated with 20 mM Tris-HCl pH 8.0, 250 mM NaCl, 20 mM imidazole and 10% glycerol. After loading, the Ni-NTA column was washed with equilibration buffer and eluted with equilibration buffer supplemented with 300 mM imidazole. After SDS-PAGE analysis, elution fractions containing SpyCas9 were pooled and dialyzed overnight at 4 °C against 20 mM HEPES pH 7.5, 500 mM KCl, and 10% glycerol. The next day, the SpyCas9 sample was concentrated to ~10 mg/mL, aliquoted, and flash-frozen in liquid nitrogen for long-term storage. HF1 SpyCas9 contains an N-terminal MBP-fusion tag. In order to remove it, Ni-NTA elution was followed by overnight TEV cleavage, and reverse Ni-NTA before proceeding with the overnight dialysis step.

For the long protocol, in order to remove any bound bacterial nucleic acids and endotoxins after elution from the Ni-NTA column, WT SpyCas9 protein was further purified using a combination of anion exchange (IEX) and heparin chromatography. The elution fractions from the Ni-NTA were diluted eightfold with 20 mM Tris-HCl pH 8.0, 250 mM NaCl, 20 mM imidazole, 1 mM DTT, and 10% glycerol (IEX equilibration buffer) and then loaded onto a 5 mL HiTrap Q HP column (Cytiva) coupled in tandem to a 1 mL HiTrap Heparin HP column (Cytiva). After washing with IEX equilibration buffer, the HiTrap Q HP column (which should bind the endotoxin molecules) was removed, and SpyCas9 protein was eluted from the HiTrap Heparin HP column in a gradient going from 250 mM NaCl to 1 M NaCl over 20

column volumes (SpyCas9 protein usually elutes ~650 mM NaCl). The elution fractions containing SpyCas9 were pooled, concentrated to ~5 mL and injected into a HiLoad 16/600 Superdex 200 pg size exclusion chromatography column (SEC) pre-equilibrated with 2 mM HEPES pH 7.5, 500 mM KCl and 10% glycerol. SpyCas9 elution fractions were pooled and concentrated to ~10–15 mg/mL. The final SpyCas9 samples were aliquoted, flash-frozen in liquid nitrogen, and stored at −80 °C until usage. The yield is usually around 7 mg of pure SpyCas9 protein from 1 L expression culture.

The identity of the SpyCas9 protein was verified by mass spectrometry, and the oligomerization state and absence of aggregates were checked by SEC-MALS and Refeyn mass photometry. The stability in the storage buffer was assessed by nano-Differential Scanning Fluorimetry (nano-DSF; $T_m$ ~ 45 °C).

## Transfection of *Dictyostelids*

The 20 nt spacer region of crRNA was custom-designed for each target using Cas designer (http://www.rgenome.net/cas-designer/) following the criteria described in Appendix Text S2, Step 1. crRNA was ordered as Alt-R CRISPR-Cas9 crRNA from IDT. For non-axenic strains, four SM5-agar plates with two spotted fronts were grown for 3-5 days (see above and Appendix Text S2, Step 4.1). Cells were collected from the feeding front, centrifuged, and washed three times in 1 mL ice-cold H50 buffer. For axenic strains, cells were grown in HL5 + FAB (Appendix Text S2, Step 3.2) to the exponential phase, harvested by centrifugation, and washed three times in 1 mL ice-cold H50 buffer (Appendix Text S2, Step 4.2). For both axenic and non-axenic strains, cells were counted using the Countess III cell counter and aliquoted to $1.25–5 × 10^6$ cells per transfection (Appendix Fig. S1A). Cells were kept on ice while preparing the RNP mix.

The RNP complex (10 µL) was prepared at RT for each transfection either with or without pre-annealing the tracrRNA and crRNA into a gRNA complex. Without pre-annealing step, reagents were mixed in the following order, by slowly adding each reagent and swirling the tip into the mixture, gently pipetting up and down once or twice after adding each reagent to a final concentration of (1) 12 µM SpyCas9 (ultrapure WT SpyCas9, unless specified otherwise), (2) 83 mM KCl, (3) 17 mM HEPES pH 7.5, (4) 22 µM Alt-R CRISPR-Cas9 crRNA, and (5) 22 µM Alt-R CRISPR-Cas9 tracrRNA (IDT) in a total volume of 10 µL. When including a pre-annealing step, crRNA and tracrRNA were mixed and incubated at 95 °C for 5 min and annealed by slow cooling. The gRNA complex was subsequently mixed with SpyCas9 at a final concentration of 15 µM (following IDT recommendation). Pre-annealing the gRNA complex improves editing efficiencies (Appendix Fig. S2) and is therefore recommended (Appendix Text S2). We specify in the figure caption when experiments were performed without including a pre-annealing step. For knock-outs, single-stranded donor oligos (IDT) were suspended at a concentration of 100 µM in nuclease-free water and added to the RNP complex. Volume was adjusted to 100 µL with H50 buffer. Oligo concentration in the final 100 µL transfection mix ranged from 0 to 4.8 µM. See Appendix Table S2 for the crRNAs and donor oligos used to produce each knock-out strain. Donor oligos under 100 nt (e.g., the donor oligo we used for knock-outs with 37 bp insertion and 28 bp homology arms) can be synthesized as regular oligos. The efficiency of these low-cost oligos is similar to high-quality

Ultramer donor oligos (Appendix Fig. S1B) and can therefore be used indistinctively.

For knock-ins, donor PCR product was PCR-amplified from pMG005 (for *mNeonGreen* reporter constructs) or pMG008 (for *mCherry* reporter constructs) (Appendix Text S1 and Appendix File S1) using custom primers (Appendix Table S4 for primer set, and Appendix Table S6 for primer sequence) with 30 to 40 bp homology arms to the CRISPR target site. For each 400 μL of PCR reaction, one Qiagen QIAquick PCR purification column was used. DNA was eluted in 22 μL nuclease-free water. The concentration of the donor PCR product was adjusted to 1 μg/μL (1.8 μM for an 899 bp PCR product consisting of mNeonGreen-P2A and 34/37 bp homology arms). Amounts between 0 and 20 μg (0–36.12 pmol) were added to the RNP mix as indicated. When indicated, a non-specific oligo (Appendix Table S6) and/or Alt-R Cas9 Electroporation Enhancer (IDT) was added to the RNP mix as well. After all the components were added, H50 buffer was slowly added to the mix at RT to have a final volume of 100 μL. When indicated, Alt-R HDR Enhancer V2 (IDT) was added to the growth media.

Once the transfection mix (containing the RNP complex, donor DNA (when indicated), enhancers (when indicated), and H50 electroporation buffer) was ready, aliquots of *Dictyostelid* cells were centrifuged and resuspended in 100 μL of transfection mix. In total, 80 μl were transferred to a pre-chilled Gene Pulser/MicroPulser Electroporation Cuvettes (Bio-Rad, Appendix Text S2) and electroporated using a Gene Pulser Xcell Microbial System with the following settings: Exponential protocol; voltage, 750 V; capacitance, 25 μF; resistance, infinite; cuvette, 1 mm; number of pulses, 1. The time constant was between 0.7 and 1 ms. Electroporated cells were kept on ice for 5 min after electroporation and added to the indicated liquid or agar growth media (Appendix Text S2, Step 4.3 and 4.7). Cells were incubated for 1–3 days at 22 °C before proceeding to flow cytometry analysis and/or sorting. Fresh media was added every 24 h.

### Flow cytometry analysis and cell sorting

For flow cytometry (Appendix Text S2, Step 5), cells were harvested from either liquid or agar media (see Appendix Text S2, Step 5.1) 2- or 3-days post-transfection in KK2-MC buffer. In total, 200 μL of resuspended cells were transferred to FACS tubes for analysis. DRAQ7 (ThermoFisher) was used as a viability dye at a 1/100 dilution. Samples were analyzed using the BD FACSymphony A3 analyzer (5 laser configuration) using BD FACSDiva software Version 9.1.2 and the following settings: FSC: 100 V, SSC: 150 V, 488-530_30: 285 V, 561-610_20: 434 V, 640-670_30: 500 V. 10.000 events were recorded at a flow rate of up to 500 events/s. Results were analyzed using FlowJo Version 10.9.0, by gating against bacterial cells (for non-axenic cultures), dead cells (DRAQ7 positive), and doublets. For the remaining single amoebal cells, we gated based on the fluorescent signal.

For sorting (Appendix Text S2, Step 6), cells were typically harvested 1- or 2-days post-transfection in KK2-MC buffer containing 2× streptomycin and penicillin (Appendix Text S2, Step 6.1). Resuspended cells were collected in a 5 mL round-bottom tube with a 35μm mesh cell strainer and sorted using BD FACSAria Fusion Flow Cytometer using BD FACSDiva software Version 8.0.1 with the following settings: FSC, 0 V; SSC, 242 V; 488–530 (30), 362 V; 561–610 (20), 539 V; 640–670 (30), 466 V; nozzle, 100 μm;

sorting target, 1 cell; flow rate up to 400 cells/s. Single live (DRAQ7 negative) cells were sorted based on fluorescence in individual wells from a 24-well plate, where each well contained 1 mL SM5 charcoal agar and a lawn of *E. coli* B/r (Appendix Text S2, Step 6.1). Sorting results were analyzed using FlowJo Version 10.9.0. Sorted cells were incubated at 22 °C for 3–5 days, until growth was observed, and then imaged using a Zeiss AxioZoom V16 (see below).

### Microscopy

For imaging slugs, clones were propagated from sorted wells to SM5 agar plates, containing 0.5% charcoal and a lawn of *E. coli* B/r, until a feeding front and consequently slugs could be detected (charcoal was added to minimize background fluorescence from the agar medium). As a control, the same procedure was followed for WT cells. Both WT and edited slugs were imaged using the Leica Stellaris 8 inverted confocal microscope with a 10 × 0.3 NA air objective. The white light laser wavelength was tuned to 488 nm with a smart intensity of 50.47%. A NF 488/561/730 Polarization filter was used to eliminate fluorescent signal from the cellulose stalk. A HyD detector was used with gain of 26.6 and a collection window of 494nm-586nm. Resolution was 1024 × 512 pixels, scan speed was 400 Hz, line averaging of 2 was used, and pixel dwell time was 1.4125 μsec. Z-stacks were taken to capture the entire slug, with a z-step size of 4.017 μm. All samples were imaged on the same day with the same acquisition settings to ensure accurate quantitative analysis. Confocal images of single *D. discoideum* cells were taken with a Nikon Ti2-E CSU-W1 spinning-disk confocal microscope with a Hamamatsu Orca FusionBT camera with a resolution of 2048 × 2048 pixels. Images were acquired either with a CFI Apo LWD 40 × 1.15 NA water immersion (MRD77410) or CFI P-Apo Ph3 100 × 1.45 NA oil (MRD31905) objective. For 40x images, cells were imaged in wide-field mode with a Lumencor Spectra III LED source and 475/28 bandpass filter (200 ms exposure time). For ×100 images, cells were imaged in spinning-disk confocal mode with 488 nm laser (37.7% intensity). Z-stacks were taken to capture entire cells, with a z-step size of 0.2 μm with 25 steps. Raw image data as well as expression profiles are provided in Appendix File S2, S3 and S4. See 'Data analysis' below for a description of our quantification methods.

Imaging of multi-well plates after cell sorting and time-lapse imaging was acquired using a Zeiss AxioZoom V16 microscope with a Plan Z ×1.0 objective and an Axiocam 506 mono camera with a 1x camera adapter. For time-lapse imaging, temperature was maintained at 22 °C using an Ibidi stage incubator (Silver Line, TC3) and a RT of 18 °C. In-well temperature was checked using the supplied test probe. Time-lapse imaging experiments were carried out in six-well plates with SM5 charcoal agar and *E. coli* B/r as food source in two rounds (see also Appendix File S5 and S6), either with *D. discoideum, D. firmibasis*, and *P. violaceum*, or with *T. lacteum, H. pallidum*, and *D. purpureum*. In each round of imaging, both fluorescent genome-edited strains and WT controls for each species were imaged. Each well was imaged at a zoom of ×1 and a 4 × 4 tile scan. Imaging was performed with fluorescence (Excelitas X-Cite Xylis LED unit, 73% power, Zeiss 38 HE GFP filter cube) with an exposure time of 1 s, two white light spotlights (Zeiss CL 9000 LED unit, 40% power) arranged from above pointing down, and transmitted light from below (Zeiss CL 9000 LED unit, 60% power). Time-lapse images (Appendix File S6) were processed in

the following way: Flatfield correction was performed using Fiji and the Biovoxxel toolbox (https://github.com/biovoxxel/Biovoxxel-Toolbox) with the first time point as template. Brightness, contrast, and gamma levels were adjusted to best show both cell spreading and aggregation. Adjustments varied between species, but were identical for control and fluorescent strains from each species (see Appendix Files S5 and S6). Tile stitching was done in ZEN 3.10 using default settings. All Appendix Files are deposited to Zenodo (https://doi.org/10.5281/zenodo.16919775).

## Quick genomic DNA (gDNA) extraction and confirmation of clones

To confirm genome edits of clones (Appendix Text S2, Step 6.3), 3–5 sori were picked from a well once fruiting bodies became visible (3–5 days after sorting transfected cells) with a small tip and submerged into a PCR tube containing 20 µL of Quick Genomic DNA extraction buffer. Cells were lysed at 56 °C for 45 min, and then heat-inactivated at 95 °C for 10 min. Primers around the editing site were designed for each construct (see Appendix Tables S2, S4 and S6 for primer lists and sequences). In all, 2 µL of gDNA were used in a 20 µL clone-confirmation PCR reaction with Taq polymerase. 2 µL of the PCR product was run in a 1% agarose gel for visualization of the results. Positive PCR reactions were purified with Qiagen QIAquick PCR Purification Kit, and the PCR fragments were eluted with 25 µl of EB. When indicated, PCR fragments were Sanger sequenced using either Eurofins TubeSeq Supreme run or Eurofins PlateSeq Kit PCR (Appendix File S7).

## gDNA extraction and Nanopore sequencing

For Nanopore sequencing, monoclonal cells were grown axenically in HL5 + FAB until reaching exponential phase ($0.8–1.2 \times 10^6$ cells/mL). For each clone, $3 \times 10^7$ cells were harvested by centrifugation at $400 \times g$ for 5 min and gDNA was extracted as previously described (Pilcher et al, 2007). Library prep and Nanopore sequencing were performed by EMBL's Genomics Core (GeneCore) Facility. In brief, DNA fragment sizes were analyzed using the Femto Pulse Systems (Agilent), and samples were chosen which had a suitable proportion of longer fragments (>10 kb). Shorter fragments were depleted from the samples using PacBio SMRTbell cleanup beads (PacBio 102-158-300) at a 3.7× ratio and selected fragments resuspended in PacBio Elution Buffer (PacBio 101-633-500). Libraries were prepared for long-read sequencing using the Ligation sequencing gDNA—Native Barcoding Kit 24 V14 (Oxford Nanopore Technologies). In all, 10 fmol of library was loaded on an R10 GridION flowcell for sequencing.

## Nanopore sequence analysis

For genome assembly, raw Nanopore reads were base-called live during sequencing using Dorado 7.4.14 (https://github.com/nanoporetech/dorado). Passed base-called nanopore reads were concatenated per sample. The *D. discoideum* AX2 *act5::mCherry* nanopore reference was assembled by combining reads from all our *D. discoideum* AX2 strains with *act5::mCherry* to maximize coverage and thereby assembly quality (see Appendix Table S3). De novo genome assembly was performed using Flye 2.9.2-b1786 (Kolmogorov et al, 2019) using the following parameter settings:

```
flye --nano-raw concatenated_reads.fastq -t 32 -o
```
assembly. Our assembly showed a 96.596% completeness compared to the RefSeq genome (GCA_000004695.1) of *D. discoideum* AX4 (Eichinger et al, 2005), as determined by QUAST v5.2.0. Differences between our *D. discoideum AX2 act5::mCherry* (MG38; Appendix Table S1) assembly and the RefSeq genome were determined using Minimap2 2.22-r1101 (Li, 2018). Except for few minor genomic rearrangements (e.g., inversions) and the *act5::mCherry* integration no markable differences were detected. Indexes for aligned fasta files were derived using samtools 1.21. Visualization was done using D-Genies (Cabanettes and Klopp, 2018), selecting Minimap2 v2.28 as the aligner and 'Many repeats' as the level of repeatedness. To examine gene edits in *mCherry* and potential off-target integration of donor oligos, the *mCherry* sequence flanking the integration site as well as 37 nt donor oligo (without homology arms) were separately queried against all raw nanopore reads using nucleotide-nucleotide BLAST 2.15.0+ (Camacho et al, 2009) using the following command line:

```
makeblastdb -in $nanopore_reads.fasta -dbtype
nucl -out $reads_with_hit.db blastn -query
[query sequence] -db $ reads_with_hit.db -outfmt
6 -out $reads_to_check.txt'.
```
All BLAST hits were subsequently examined manually by aligning the positive reads against our de novo genome assembly using SnapGene 8.0.2 (www.snapgene.com; option 'Aligned to Reference DNA Sequence'). In addition, all BLAST hits were also exported to a fasta file and automatically aligned against our reference genome using Minimap2 2.22-r1101. The resulting BAM files were furthermore filtered (`-q 30` and `-F 0×900`) and sorted using Samtools 1.21, making it possible to visualize the alignment against our reference genome using Integrative Genomics Viewer (IGV) (Appendix Table S3). No incongruencies were detected, meaning that all *mCherry* genes had the expected edit and all donor oligos were integrated in the expected *mCherry* site (i.e., no off-target integration; Appendix Table S3). The Nanopore sequence analysis was performed on EMBL's HPC Cluster (https://doi.org/10.5281/zenodo.12785829).

All sequencing-related files, as listed in Appendix Table S3, are deposited to Zenodo (https://doi.org/10.5281/zenodo.16919775) with folder (A) containing all passed base-called Nanopore reads, (B) containing our annotated de novo *D. discoideum* AX2 *act5::mCherry* genome assembly (our reference genome), (C) alignment of all reads against our reference genome (i.e., original BAM files), (D) alignment of all reads containing *mCherry* against our reference genome (see Appendix Table S3), (E) alignment of all reads containing donor oligo against our reference genome (see Appendix Table S3).

## Data analysis

Knock-out and knock-in percentages were quantified from flow cytometry data (Appendix File S8) using FlowJo Version 10.9.0. Maximum knock-out or knock-in efficiencies were assessed by fitting a sigmoidal curve as a function of repair template concentration (i.e., donor oligo or donor PCR product) using a non-linear least square fitting (nls function in R v4.2.2): $\% = max/\left(1 + e^{scaling \bullet (mid - [donor])}\right)$. Estimated mean and standard error for the maximal editing efficiency (*max*) and inflection point (*mid*) are provided in Appendix Data S1 for each figure. Sigmoidal fits are only shown when estimated parameter values are significant

($P < 0.05$). Plots in other figures were generated in R (v4.2.2) as well. Microscopy images were processed using either Fiji (v1.53t) or Matlab R2024a. For examining expression profiles of slugs, a maximum projection was acquired for each z-stack, and then a 20-µm-wide spline was manually drawn from a slug's anterior to posterior. Using these splines, expression profiles were quantified using a 20µm-rolling window from anterior to posterior. All raw image data and expression profiles are provided in Appendix File S3. For analyzing expression profiles in cells with *act5* expression reporters, cells were first segmented based on their maximum projection. Then, the distance of pixels to the cell periphery was measured using `bwdist` (function in Matlab) to project expression profiles along the radial axis of cells. For each cell, expression intensities are projected using a 2µm-rolling window from the cell periphery to the center. Maximum projections, pixel intensities, and distances are provided in Appendix File S4. The phylogenetic tree with *Dictyostelid* species was constructed using RaxML, based on 18S rDNA sequence alignment from Sheikh and colleagues (Sheikh et al, 2018), using the following parameter settings: `raxmlHPC -f d -m GTRGAMMA -n TREE -s 1-s2.0-S1434461017300925-mmc13.fasta -# 100 -b 1 -p 1`. All files (Appendix File S1-S8) and data (Appendix Data S1) are deposited to Zenodo (https://doi.org/10.5281/zenodo.16919775).

## Data availability

All source data of this paper (Appendix Data S1), including plasmid sequences (Appendix File S1), image data and expression profiles (Appendix Files S2–S6), sanger sequencing data (Appendix File S7), flow cytometry data (Appendix File S8) and nanopore data (listed in Appendix Table S3) can be accessed through Zenodo under accession number 16919775 (https://doi.org/10.5281/zenodo.16919775).

The source data of this paper are collected in the following database record: biostudies:S-SCDT-10_1038-S44320-025-00180-8.

## Peer review information

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

## Acknowledgements

We thank everybody from the van Gestel group for feedback and suggestions. We particularly thank Öykü Bozalioğlu for establishing efficient electroporation conditions, Haohan Zhang for testing different isolation methods, and Vanessa Stürmer for her support in taking microscopy images. We thank EMBL's Advanced Light Microscopy Facility (ALMF), Flow Cytometry Core Facility, and GeneCore Facility for their support. We thank Peggy Paschke for her strains and plasmids as well as her generous advice on genome editing methods. We thank Elizabeth Ostrowski and Pauline Schaap for their advice on culturing diverse *Dictyostelid* species. We thank Allyson Sgro, Emily Hager, John Durel, and Elizabeth Ostrowski for their feedback on the manuscript. We thank EMBL for financial support. AP also received support from iXcore – iXlife – iXblue foundation. JvG received support from the European Union (ERC, CO-PP, 101116560). Views and opinions expressed are however those of the author(s) only and do not necessarily reflect those of the European Union or the European Research Council Executive Agency. Neither the European Union nor the granting authority can be held responsible for them.

## Author contributions

**Mireia Garriga-Canut**: Conceptualization; Formal analysis; Validation; Investigation; Visualization; Methodology; Writing—original draft; Writing—review and editing. **Nikki Cannon**: Formal analysis; Investigation; Visualization; Writing—original draft. **Matt Benton**: Formal analysis; Investigation; Visualization; Writing—original draft. **Andrea Zanon**: Investigation; Visualization; Writing—original draft. **Samuel T Horsfield**: Supervision. **Jacob Scheurich**: Resources; Investigation. **Kim Remans**: Resources; Supervision. **John Lees**: Supervision. **Alexandre Paix**: Conceptualization; Supervision; Writing—original draft; Writing—review and editing. **Jordi van Gestel**: Conceptualization; Data curation; Formal analysis; Supervision; Funding

acquisition; Visualization; Methodology; Writing—original draft; Project administration; Writing—review and editing.

Source data underlying figure panels in this paper may have individual authorship assigned. Where available, figure panel/source data authorship is listed in the following database record: biostudies:S-SCDT-10_1038-S44320-025-00180-8.

## Funding

## Disclosure and competing interests statement
The authors declare no competing interests.

