## [Peer Review File · Molecular Systems Biology]

Unlocking CRISPR-Cas9 editing for widely diverse Dictyostelid species

Mireia Garriga-Canut, Nikki Cannon, Matt Benton, Andrea Zanon, Samuel Horsfield, Jacob Scheurich, Kim Remans, John Lees, Alexandre Paix, and Jordi van Gestel

Corresponding author(s): Jordi van Gestel (jordi.vangestel@embl.de) , Alexandre Paix (alexandre.paix@tuebingen.mpg.de)

Review Timeline:

Submission Date:	3rd May 25
Editorial Decision:	15th Jun 25
Revision Received:	11th Sep 25
Editorial Decision:	20th Oct 25
Revision Received:	28th Oct 25
Editorial Decision:	8th Nov 25
Revision Received:	21st Nov 25
Accepted:	24th Nov 25

Editor: Yehu Moran

Transaction Report:

15th Jun 2025

Manuscript Number: MSB-2025-13091
Title: Unlocking CRISPR-Cas9 editing for widely diverse Dictyostelid species
Author: Mireia Garriga-Canut
Nikki Cannon
Matt Benton
Andrea Zanon
Samual Horsfield
Jacob Scheurich
Kim Remans
John Lees
Alexandre Paix
Jordi van Gestel

Dear Dr. van Gestel,

Thank you again for submitting your work to Molecular Systems Biology. We have now heard back from the three referees who agreed to evaluate your manuscript. As you will see from the reports below, the referees find the topic of your study interesting and relevant. They raise, however, multiple concerns that require your attention.

When you resubmit your manuscript, please download our CHECKLIST (<https://bit.ly/EMBOPressAuthorChecklist>) and include the completed form in your submission.

Please note that the Author Checklist will be published alongside the paper as part of the transparent process (<https://www.embopress.org/page/journal/17444292/authorguide#transparentprocess>).

In light of these three positive reviews I invite you to submit a revised version of your manuscript. Please attach a covering letter giving details of the way in which you have handled each of the points raised by the referees. A revised manuscript will be once again subject to peer review and you probably understand that we cannot give you guarantee at this stage that the eventual outcome will be favorable, just like with any peer review process. If you have any specific concerns or questions regarding the process or you would like to receive any feedback regarding your revision plan, you are welcome to contact me at my email address y.moran@molsystbiol.org

Yours sincerely,

Yehu Moran
Academic Editor
Molecular Systems Biology

We realize that it is difficult to revise to a specific deadline. In the interest of protecting the conceptual advance provided by the work, we recommend a revision within 3 months (13th Sep 2025). Please discuss the revision progress ahead of this time with the editor if you require more time to complete the revisions. Use the link below to submit your revision:

IMPORTANT: When you send your revision, we will require the following items:

1. the manuscript text in LaTeX, RTF or MS Word format
 2. a letter with a detailed description of the changes made in response to the referees. Please specify clearly the exact places in the text (pages and paragraphs) where each change has been made in response to each specific comment given
 3. three to four 'bullet points' highlighting the main findings of your study
 4. a short 'blurb' text summarizing in two sentences the study (max. 250 characters)
 5. a 'thumbnail image' (550px width and max 400px height, Illustrator, PowerPoint or jpeg format), which can be used as 'visual title' for the synopsis section of your paper.
 6. Please include an author contributions statement after the Acknowledgements section (see <https://www.embopress.org/page/journal/17444292/authorguide>)
 7. Please complete the CHECKLIST available at (<https://bit.ly/EMBOPressAuthorChecklist>).
- Please note that the Author Checklist will be published alongside the paper as part of the transparent process (<https://www.embopress.org/page/journal/17444292/authorguide#transparentprocess>).

See also figure legend guidelines: <https://www.embopress.org/page/journal/17444292/authorguide#figureformat>

9. Please note that corresponding authors are required to supply an ORCID ID for their name upon submission of a revised manuscript (EMBO Press signed a joint statement to encourage ORCID adoption).

(<https://www.embopress.org/page/journal/17444292/authorguide#editorialprocess>)

Currently, our records indicate that the ORCID for your account is 0000-0001-5598-5239.

Link Not Available

11. Include a Reagents and Tools Table as part of the Methods section, which can be downloaded from our author guidelines (<https://www.embopress.org/page/journal/17444292/authorguide#structuredmethods>)

*** PLEASE NOTE *** As part of the EMBO Press transparent editorial process initiative (see our Editorial at <https://dx.doi.org/10.1038/msb.2010.72>), Molecular Systems Biology publishes online a Review Process File with each accepted manuscripts. This file will be published in conjunction with your paper and will include the anonymous referee reports, your point-by-point response and all pertinent correspondence relating to the manuscript. If you do NOT want this File to be published, please inform the editorial office at contact@molsystbiol.org within 14 days upon receipt of the present letter.

Reviewer #1:

In this study the authors developed a new CRISPR-Cas9 approach for Dictyostelids. The method is cloning free and they convincingly show that genome editing is highly efficient, fast and can be used for phylogenetically distant Dictyostelids (although with lower efficiency). This approach can thus have a major impact on the field and may allow broad scale interrogations across Dictyostelid families. The study is of high quality and can be accepted in the current form. However, it would even further improve if they can show more examples of their approach: is efficient of KO of other genes as efficient as shown for mCherry? Can their approach be used to efficiently generate KO of the other Dictyostelids?

Reviewer #2:

This study presents a novel method for genome editing across various species of Dictyostelid social amoebae. While *Dictyostelium discoideum* has long been the primary model organism, most other Dictyostelid species have remained genetically inaccessible due to the lack of efficient manipulation techniques. To address this limitation, the authors developed a new CRISPR-Cas9 protocol that is compatible with both axenic and non-axenic growth conditions. By delivering ribonucleoprotein (RNP) complexes of SpyCas9 and guide RNAs directly into cells via electroporation, and utilising donor oligonucleotides or PCR-generated templates for homology-directed repair, they achieved high editing efficiencies. Furthermore, single-cell sorting enabled the rapid and selective isolation of edited clones as early as one day post-transfection, particularly in cases where fluorescent proteins were knocked in. Notably, the authors provided clear quantitative evaluations of key factors influencing editing efficiency, such as homology arm length, insertion size, and donor concentration. This level of optimisation, which has been previously lacking in the field, offers valuable practical guidance for future applications.

I am convinced by the strength of the data and the clarity of the experimental design that the key conclusions are well supported. This work represents a major technical advancement, expanding the toolkit for genetic manipulation across Dictyostelid species of ecological, developmental, and evolutionary importance. Compared to previous methods that were limited to a few model strains, this approach is considerably more versatile, rapid, and accessible. It will be of interest not only to researchers studying *Dictyostelium* biology but also to those in the broader fields of protist genetics, microbial evolution, and developmental systems. I believe that the authors' significant achievements hold substantial value for publication, provided that the following concerns are adequately addressed.

Major points:

1. L25, L750: Although the reported maximum efficiencies (~80% for knock-out and ~30% for knock-in) are encouraging, these reflect outcomes under optimal conditions at the *act5* locus with a mCherry insert. As shown in Figures 5, 6, S5, S10, etc.,

- efficiencies can vary considerably depending on the target gene, insertion sequence, and specific site. To avoid giving a potentially misleading impression, the authors should clarify that these figures represent maximal efficiencies. Additionally, providing a brief summary of the observed range of efficiencies across different targets would offer a more balanced perspective on the method's overall performance and assist researchers in evaluating its applicability to their own experimental systems.
2. L412: In the section titled "RNP complex mediates high knock-out efficiencies in *D. discoideum*," the authors highlight the potential of their RNP-based method; however, the knock-out efficiency achieved with RNP alone is reported to be around 5%. In contrast, Asano et al. reported knock-out efficiencies exceeding 95% using SpCas9-NG-a slightly less efficient variant than SpCas9-in a transient plasmid-based system involving one to two days of drug treatment. To clearly demonstrate the added value of the RNP-based approach, a direct comparison with such plasmid-based transient systems would be informative.
 3. L530-532 & Figure 3: The non-axenic strains used for transfection are described as being collected from the periphery of bacterial plaques. The authors compare knock-out efficiencies under different bacterial densities, which seem to reflect the conditions after transfection. It is not entirely clear why the bacterial density post-transfection was selected as the variable of interest, or what effect the authors expected it would have on the editing process. As such, the rationale for this comparison would benefit from further clarification to help readers understand the purpose and interpretation of the results.
 4. The approach using single-cell sorting assumes that non-fluorescent cells represent knock-outs. However, in the sorting experiment described in Figure 4C, only 35 out of 48 wells showed growth (line 569), indicating that around 27% of the sorted cells failed to proliferate. The reason for this lack of growth is not addressed, but it could have a considerable impact on the overall knock-out efficiency reported in the study.
 5. L629-638: Knock-in efficiency is inferred solely from the presence or absence of fluorescence. However, fluorescence alone does not confirm that integration has occurred at the correct genomic locus. True knock-in events should be defined by precise insertion at the intended site, whereas random integration or genomic rearrangements could also give rise to fluorescent signals. This issue is particularly relevant in the case of H2Bv3, a major histone H2B variant, where accurate targeting may be challenging-similar to the known difficulties in generating knock-out and knock-in of major histones such as H3a. Therefore, PCR amplification of the targeted genomic region, followed by Sanger sequencing, is necessary to verify precise integration. Similarly, for *act5*, confirmation by PCR was not demonstrated, even though this was addressed elsewhere in the manuscript (e.g., lines 600-601).
 6. L682-686: Previous research by Muramoto and colleagues demonstrated the successful simultaneous editing of up to five genes, while they developed a convenient expression vector that enables dual-site editing. If the authors wish to highlight the efficiency or scalability of their method for multiplexed editing, it would be beneficial to compare their findings with these earlier studies. Furthermore, since the present approach involves the co-delivery of multiple donor oligonucleotides, the potential for interference among them should be considered, particularly when targeting more than two loci.
 7. While the advantages of the method are well described in the Discussion, its limitations are not sufficiently addressed. In particular, although the addition of donor oligonucleotides clearly increases knock-out efficiency, it may also decrease the likelihood of isolating in-frame or frameshift mutations in essential genes, as noted by You et al. 2020. For example, in donor-free genome editing targeting essential genes such as "tor", knock-out mutants could not be obtained; however, in-frame and frameshift variants were successfully isolated and analysed (Li et al., 2016; Ogasawara et al., 2022). Therefore, it would be beneficial for the authors to consider and briefly discuss the potential limitations of combining RNP with donor oligos, particularly in the context of editing essential genes.

Minor points:

1. Introduction: The authors should carefully distinguish between traditional gene manipulation via homologous recombination and CRISPR-based genome editing. For example, line 49 refers to conventional methods as "genome editing," while lines 66-67 describe CRISPR specifically. Since Schaap and colleagues used traditional recombination rather than CRISPR, careful use of terminology would help avoid potential confusion. Moreover, it would be helpful for the Introduction to more clearly acknowledge that previous comparative analyses were conducted using strains generated by traditional knock-out methods. Highlighting known limitations in *P. pallidum*, such as the low efficiency of gene disruption and the restricted range of available selection markers, would further underscore the value and advantages of the approach presented in this study.
2. The Methods section is somewhat redundant, as detailed protocols are also provided in the Supplementary file. It may improve readability to restructure this section by retaining only the most essential information in the main text.
3. The use of cell sorting for clonal isolation is a valuable aspect of this study, particularly effective when fluorescent proteins are expressed. However, in the case of standard gene knock-outs without a fluorescent marker, sorting serves primarily as a general cloning tool. To address a broader range of applications, it would be helpful for the authors to also consider and compare traditional plaque-based cloning on bacterial lawns.
4. L190: The *act5* target, it appears that no such spacer without potential off-target sites could be identified. It would therefore be more accurate to revise this statement to reflect the actual design constraints.
5. L210: (4)  (5)
6. L212: The manuscript describes two approaches for preparing the RNP complex, with crRNA and tracrRNA either added separately at a final concentration of 22 μ M, or pre-annealed and used at 15 μ M. However, the reason for this difference in concentration is not clearly explained. Additionally, it is unclear which of these two methods was used in each experiment.
7. L305: The species names are conventional names, whereas the rest part of the manuscript uses the nomenclatures proposed in ref. 5.
8. L423-426: The use of the RNP complex in *Dictyostelium* is a key factor enabling a selection-free genome editing approach. While this point is mentioned in parts of the manuscript, it would benefit from being stated more clearly and emphasised, as it represents a significant advantage of the method.

9. In Figure S2, knock-out appears to be achieved using a donor oligo, whereas lines 433-437 and Figure S3A-B seem to describe an NHEJ-based approach. As this could lead to confusion, it should be clarified which strategy was used in each case.
10. L434: It is unclear how many clones were analysed in total, and whether the results shown in Figure S3 represent all analysed clones or only selected successful ones. If the labels such as "MC1_" represent clone names, the numbering in Figure S3C seems inconsistent, as "3" appears twice.
11. L460: The data indicate that 56 bp arms actually yielded higher knock-out efficiency. It would be advisable to revise the wording to reflect this more precisely.
12. L486-487: The term "PCR template" should be corrected to "PCR product".
13. L528: Schaap and colleagues previously incubated non-axenic strains in HL5 medium for one day prior to transformation to reduce nuclease activity, which may be elevated during bacterial growth. Since this step appears to have been omitted in the present study, it would be beneficial to discuss whether this omission could have affected transformation efficiency or genome editing outcomes.
14. L528, L531: The sentences describe transfections done with bacterially grown cells, but not transfections done in the presence of bacteria.
15. Figure 4, L548-576: It may seem unusual for a control experiment related to sorting to be presented as a main figure. The novelty and significance of this figure could be more explicitly stated.
16. Fig5D: In the act5::mNeonGreen reporter, fluorescence appears diminished at the anterior region of the slug. It would be beneficial to clarify whether this apparent non-uniformity is a result of projection artifacts or if it reflects a known characteristic of mNeonGreen, such as the uneven fluorescence observed during late developmental stages (Hashimura et al., 2024).
17. L708: ref 49 & 63 describe the generation of knock-out cells via homologous recombination, rather than knock-in approaches.
18. L710: The reported knock-in efficiency in *D. discoideum* (16%) is somewhat unclear, as it is not specified which experiment or figure this value is derived from.
19. L712: ref35 reports optimisation of the electroporation buffer specifically for cells cultured with bacteria. In the present study, the H50 buffer was used, but it would be helpful to include a brief discussion of this choice and whether it may have influenced transfection efficiency.
20. Figure 7: Given that *D. discoideum* possesses over 20 actin genes, and genome analysis indicates that other Dictyostelids also have multiple actin isoforms, it is essential to verify that integration has occurred at the intended locus. Referring to the outcome as a "knock-in" without such confirmation risks being misleading.
21. L749: The statement that cell stocks can be prepared "a few days later" may somewhat overstate the general applicability of the approach. While this timeline may be feasible for fluorescence-based control experiments, confirming knock-out or knock-in clones through genome sequencing would typically require more time.
22. Figure S10C: The sum of the displayed percentages is 101%.

Reviewer #3:

Summary

In this exciting study, the authors developed a method to rapidly generate CRISPR mutants in axenic, non-axenic, and phylogenetically distant Dictyostelid species. Their approach was to deliver the CRISPR/Cas 9 ribonucleotide complex (RNP) into the amoeba via electroporation instead of using all-in-one CRISPR plasmids. The RNP delivery circumvented the need for cloning unlike with the CRISPR vector approach, shortening the time to generate mutants. Additionally, the authors used single cell sorting ~2-3 days after transfection to bypass time-intensive selection and screening of polyclonal populations and generating monoclonal populations by limiting dilutions.

As proof of concept, the authors optimized the workflow and parameters in axenically-grown AX2 cells. Then, they performed genome editing with amoeba in non-axenic growth conditions. They made clever use of fluorescent strains to perform knock-outs and knock-ins with repair templates and used flow cytometry to provide quantitative data for genomic editing efficiencies. Next, the authors edited multiple genes simultaneously under axenic conditions. Finally, the authors generated knock-outs of 6 diverged Dictyostelids, including 3 previously intractable species. By eliminating the cloning and selection steps, as well as the reliance on limited selectable markers, this study dramatically increases the genetic tractability of this model organism, as well as several other Dictyostelium species.

General remarks

There are currently a very limited set of tools for genomic editing of non-axenic Dictyostelium. This manuscript identifies clear gaps in this field and tackles them directly through technical innovation. The authors were extremely thorough, covering knock ins, knock outs, fluorescent reporters, endogenous fusion proteins, rates of NHEJ vs HR, effective ways to isolate clones, and combinations of mutations. In line with their thoroughness, the authors included the nanopore sequencing and assembly of the AX2 act5::mCherry strain to look for off-target edits. This is above and beyond where the field has been to this point. This work would expand the toolset for genomic editing of Dictyostelium species, including non-axenic strains, and has the potential to become a new guidebook for researchers in the field. Additionally, the detailed information in the methods and in the supplement is very well written and will help other labs use this method successfully. This manuscript will interest most researchers working with Dictyostelium as these methods have downstream applications to many if not all facets of Dictyostelium biology. I believe that the data shown supports most of the conclusions stated in the text. A few changes to the text and data presented could strengthen the manuscript even further (suggested below).

Major points

1. The authors were able to get purified SpyCas9 from their own institution's facility. To enable broader uptake of this method, the authors do describe a method for purification, but many Dictyostelium labs do not have this set up available. It is also disappointing that many of the commercial SpyCas9s perform so poorly in Fig S5.
 - 1a. Do the authors have any suggestions (or data) about optimizing these protocols with commercial SpyCas9 sources?
 - 1b. Additionally, as this is likely the most expensive part of the protocol for most researchers, have the authors titrated how much SpyCas9 is needed in these reactions?
 - 1c. Is it possible to successfully perform more complex genomic editing (as shown in Fig 5) with commercial Cas9 from Thermo? Although the edit shown are impressive, a more thorough investigation of the commercial proteins would help other researchers to know which of the techniques presented here would plausibly work in another lab.
2. The authors conclude that their system can be used to successfully generate double knock-ins and simultaneous knock-in and knock-out mutations. I agree that the single knock-out rates are very high, but I encourage the authors to moderate their language for the other methods, as the rates are much lower. Additionally, the following edits would provide clarification and further support for the authors conclusions:
 - 2a. The methods used in Figure S10 are currently unclear. A targeting and outcome schematic similar to Figure 2A would be very helpful for clarification.
 - 2b. Because *ecmA* is not expressed in vegetative cells, we cannot assess its editing in S10A and S10C. To appreciate the changes in fluorescence resulting from genomic editing, it would help to see flow cytometry data (or similar) from disaggregated slugs for similar conditions as S10A-C. Alternatively, sequencing data could be used to assess the editing frequency in the *act5* and *ecmA* loci.
 - 2c. Despite concluding that successful mCherry knock-out occurred, panels S10D-F show that mCherry cells are still very bright (at 10^4 after knock-out), this is at odds with the "no fluorescent" labelling in the figures. Please explain. Is it possible that the labelled actin has not turned over at the time of data collection?
3. The authors conclude that they generated mutants of 6 diverged Dictyostelids without major impacts to growth and development. However, not much raw data was shown for this section of the manuscript, unlike all other sections. To strengthen the conclusions in this section, I recommend:
 - 3a. including the flow cytometry data for the strains and the images of the plates after sorting (or cell count data to assess growth).
 - 3b. including data such as images of non/mock electroporated cells at the same timepoint as the images shown in 5B to illustrate whether development or growth has been affected.
 - 3c. explaining why false color was used as a measure of fluorescence in panel 5B and not in panels 4C-D and 5A-B.

Minor points

1. Line 52- the author's name was misspelled, it is Muramoto.
2. Line 55- Muramoto's system can be used with 2 guides depending on the vector.
3. Line 69-Paschke et al should be cited here as this is one existing sample of a CRISPR system for non-axenic Dicty.
4. Line 196-what is the FAB additive to the HL5? What is its concentration?
5. Methods text has live and dead cells as DRAQ7 negative. I believe only live cells would be negative for this stain.
6. For panel 1D-use open or different symbols to clarify/differentiate this panel from the others.
7. The X axis labelling for 6A is unclear, explicitly stating N or C tag would be more clear.
8. Line 504-no citations or statistics to demonstrate that this is the highest ever recorded genetic editing without selective markers. Please add citations and include what the recorded efficiencies were in the other papers, this could be in a table format. Relatedly, knock In efficiencies of ~15-20% are labelled as high, please expand and discuss why this is with relevant comparisons and citations.
9. Panel 5E does not show the images of the wells although they were referenced in the text. This can be addressed by including the images of the wells or rewriting of the sentence in the results to indicate that 5E shows one representative image of a slug.

Reviewer 1

1.1. In this study the authors developed a new CRISPR-Cas9 approach for *Dictyostelids*. The method is cloning free and they convincingly show that genome editing is highly efficient, fast and can be used for phylogenetically distant *Dictyostelids* (although with lower efficiency). This approach can thus have a major impact on the field and may allow broad scale interrogations accros *Dictyostelid* families. The study is of high quality and can be accepted in the current form. However, it would even further improve if they can show more examples of their approach: is efficient of KO of other genes as efficient as shown for mCherry? Can their approach be used to efficiently generate KO of the other *Dictyostelids*?

We thank the reviewer for their constructive feedback. The reported efficiencies are indeed optimal efficiencies and can vary substantially between CRISPR-Cas9 targets and loci (e.g., Fig. S6). We now revised part of our abstract and discussion to highlight that we talk about optimal editing efficiencies (Lines 30-32, 787-792). Although we optimized our protocol by quantifying both knock-out and knock-in efficiencies at the *act5* locus, for knock-ins we already demonstrated that we can target other loci as well (e.g., *ecmA* and *H2Bv3*). For new experiments, we even obtained knock-in efficiencies of 40% at the *ecmA* locus that can be increased to almost 60% when performing double knock-ins (Figure S15). In addition, we now also performed additional knock-out experiments at the *pkaC* locus with three distinct CRISPR-Cas9 targets (Lines 603-613, Figure S12). Although these show lower efficiencies (2-20%) than those in *mCherry*, we managed to get the desired knock-outs for all three targets within a single transfection only. Altogether these results highlight the effectivity of our methods across loci. Since the knock-in efficiencies for other *Dictyostelid* species are still far lower than in *D. discoideum*, we can only isolate mutants through a sorting approach, but we are confident that we could improve this in the future.

Reviewer 2

This study presents a novel method for genome editing across various species of *Dictyostelid* social amoebae. While *Dictyostelium discoideum* has long been the primary model organism, most other *Dictyostelid* species have remained genetically inaccessible due to the lack of efficient manipulation techniques. To address this limitation, the authors developed a new CRISPR-Cas9 protocol that is compatible with both axenic and non-axenic growth conditions. By delivering ribonucleoprotein (RNP) complexes of SpyCas9 and guide RNAs directly into cells via electroporation, and utilising donor oligonucleotides or PCR-generated templates for homology-directed repair, they achieved high editing efficiencies. Furthermore, single-cell sorting enabled the rapid and selective isolation of edited clones as early as one day post-transfection, particularly in cases where fluorescent proteins were knocked in. Notably, the authors provided clear quantitative evaluations of key factors influencing editing efficiency, such as homology arm length, insertion size, and donor concentration. This level of optimisation, which has been previously lacking in the field, offers valuable practical guidance for future applications.

I am convinced by the strength of the data and the clarity of the experimental design that the key conclusions are well supported. This work represents a major technical advancement, expanding the toolkit for genetic manipulation across *Dictyostelid* species of ecological, developmental, and evolutionary importance. Compared to previous methods that were limited to a few model strains, this approach is considerably more versatile, rapid, and accessible. It will be of interest not only to researchers studying *Dictyostelium* biology but

also to those in the broader fields of protist genetics, microbial evolution, and developmental systems. I believe that the authors' significant achievements hold substantial value for publication, provided that the following concerns are adequately addressed.

We appreciate this positive feedback and have done our best to reciprocate their constructive feedback.

Major points

2.1. L25, L750: Although the reported maximum efficiencies (~80% for knock-out and ~30% for knock-in) are encouraging, these reflect outcomes under optimal conditions at the *act5* locus with a *mCherry* insert. As shown in Figures 5, 6, S5, S10, etc., efficiencies can vary considerably depending on the target gene, insertion sequence, and specific site. To avoid giving a potentially misleading impression, the authors should clarify that these figures represent maximal efficiencies. Additionally, providing a brief summary of the observed range of efficiencies across different targets would offer a more balanced perspective on the method's overall performance and assist researchers in evaluating its applicability to their own experimental systems.

We thank the reviewer for raising this important point. We now clarify throughout the manuscript (e.g., Lines 30-32 and 787-792) that we are referring to maximum knock-out and knock-in efficiencies. We would also like to highlight there are three distinct reasons why we sometimes have lower efficiencies: (1) there is some batch-to-batch variation across experiments; (2) some CRISPR targets provide consistently lower editing efficiencies, even under optimal conditions; and (3) for some experiments we did not use optimal conditions. For instance, not in all experiments we included a pre-annealing step between the crRNA and tracrRNA, which generally improves editing efficiencies (see also Figure S2). Given that we took a comparative approach in optimizing our protocol, these deviations from the optimal protocol do not affect our conclusions.

Rather than providing a summary with all results, which gets relatively complicated because of the many different conditions we explored, we now highlight in figure captions whenever we deviated from the optimal protocol as specified in Text S2 in the SI. With the many additional experiments performed as part of this revision, we believe that the reader gets a realistic impression about the editing efficiencies that can be achieved with our protocol.

2.2. L412: In the section titled "RNP complex mediates high knock-out efficiencies in *D. discoideum*," the authors highlight the potential of their RNP-based method; however, the knock-out efficiency achieved with RNP alone is reported to be around 5%. In contrast, Asano et al. reported knock-out efficiencies exceeding 95% using SpCas9-NG-a slightly less efficient variant than SpCas9-in a transient plasmid-based system involving one to two days of drug treatment. To clearly demonstrate the added value of the RNP-based approach, a direct comparison with such plasmid-based transient systems would be informative.

The reported >95% knock-in efficiencies for the plasmid-based CRISPR method are the product of three types of efficiencies: (1) transfection efficiency, (2) editing efficiency, and (3) antibiotic selection efficiency. The efficiencies we report only rely on the transfection and editing efficiencies, a direct comparison between our RNP-based protocol and previous plasmid-based protocols is therefore complicated and in part depends on the research aims.

When aiming to generate a single knock-in mutant in an axenic strain, for which an effective antibiotic marker is available, high selection efficiencies can largely compensate for low transfection and editing efficiencies. However, when targeting non-axenic strains for which effective antibiotic markers are lacking, this is already more difficult. Similarly, selection cannot compensate when aiming to generate a complex knock-in library using a single transfection, for which you rely a high number of edited cells, requiring both high transfection and editing efficiencies. As such, we would argue that our selection-free RNP-based protocol complements selection-based methods, by (1) saving time (no need of cloning or selection), (2) allowing for high editing efficiencies without selective marker and (3) supporting editing in non-axenic strains/species. We now better highlight these unique benefits of our approach in the Discussion (Paragraphs starting with Line 805, 821 and 839) and refined some of the comparisons we made ourselves before (e.g., Lines 503-505).

2.3. L530-532 & Figure 3: The non-axenic strains used for transfection are described as being collected from the periphery of bacterial plaques. The authors compare knock-out efficiencies under different bacterial densities, which seem to reflect the conditions after transfection. It is not entirely clear why the bacterial density post-transfection was selected as the variable of interest, or what effect the authors expected it would have on the editing process. As such, the rationale for this comparison would benefit from further clarification to help readers understand the purpose and interpretation of the results.

We thank the reviewer for their comment and realized that this section of the manuscript was somewhat confusing, so we now extensively revised the text (Lines 522-550) and performed additional experiments to better dissect the relative impact of non-axenic growth before and after transfection (Figure 3). In our original data we pre-cultured AX2 axenically and NC4 on bacteria, transfected cells and recovered them with or without bacteria. We obtained the somewhat surprising result that non-axenic growth after transfection lowers editing efficiencies. We now partly redid these experiments and, in addition, also pre-cultured AX2 with bacteria (Figure 3 and Lines 526-538). Non-axenic growth before transfection provides high editing efficiencies for both AX2 and NC4. After transfection, non-axenic growth decreases editing efficiencies (as observed before), suggesting that edited and non-edited cells are in somewhat different physiological states and therefore show heterogeneous recovery (Lines 532-534). It is unclear what explains this differential recovery, but from all non-axenic growth conditions we show that recovery on SM5 agar plates with *E. coli* B/r provides editing efficiencies that are similar to those for axenic conditions (HL5). As such, our RNP-based genome editing protocol is effective when editing non-axenic strains.

2.4. The approach using single-cell sorting assumes that non-fluorescent cells represent knock-outs. However, in the sorting experiment described in Figure 4C, only 35 out of 48 wells showed growth (line 569), indicating that around 27% of the sorted cells failed to proliferate. The reason for this lack of growth is not addressed, but it could have a considerable impact on the overall knock-out efficiency reported in the study.

For most of our optimization experiments (e.g., Figure 1 and 2) we did not sort cells but rather quantified editing efficiencies using flow cytometry. In these experiments, we directly controlled for cell death using DRAQ7 staining and always included control transfections without crRNA. We are therefore confident that DRAQ7-negative non-fluorescent cells are both viable and have the expected knock-out, as confirmed by Sanger sequencing (Figure

S3). This is furthermore supported by the gradual decline of fluorescent signal in knock-out cells following their proliferation (Figure S4) and by the fact that non-fluorescent cells are not observed when editing other loci in an *mCherry* background strain (Figure S16B and Figure S17C).

In sorting experiments, we expect that cells sometimes lyse due to shear stress, which is a particularly high in sorters due to the narrow sorting nozzle (in contrast to flow cytometers). We now mention this as well in Lines 584-585. As the recovery rates after sorting are similar between experiments with and without editing, there are no indications that sorting itself would bias our estimated editing efficiencies. This is also corroborated by a new knock-out experiment (Figure S16), where we directly sorted non-fluorescent *mCherry*-knockout cells and get similar recovery rates as for the other sorting experiments. We also confirmed that the sorted *mCherry* knock-outs have the expected donor oligo integration through both PCR and Sanger sequencing (Figure S16 and File S7).

2.5. L629-638: Knock-in efficiency is inferred solely from the presence or absence of fluorescence. However, fluorescence alone does not confirm that integration has occurred at the correct genomic locus. True knock-in events should be defined by precise insertion at the intended site, whereas random integration or genomic rearrangements could also give rise to fluorescent signals. This issue is particularly relevant in the case of H2Bv3, a major histone H2B variant, where accurate targeting may be challenging-similar to the known difficulties in generating knock-out and knock-in of major histones such as H3a. Therefore, PCR amplification of the targeted genomic region, followed by Sanger sequencing, is necessary to verify precise integration. Similarly, for *act5*, confirmation by PCR was not demonstrated, even though this was addressed elsewhere in the manuscript (e.g., lines 600-601).

We appreciate the concerns of the reviewer and the broader need for ruling out potential cross-integrations when performing complex genome edits (see also next comment). We therefore implemented the following two changes:

(1) In addition to the Sanger and Nanopore data that were already included before (Figure S3 and S5), we now added Sanger sequencing results for >100 additional isolates (see Table S6 and File S7). A large fraction of them was acquired as part of new experiments, where we did extensive PCR validations (Figure S10, S12, S15, and S16). Altogether, these new results corroborate our previous findings and show correct integrations of repair templates. For the H2Bv3 experiment specifically, we only performed the flow cytometry experiment and examined the transfected cells using microscopy directly. We have not isolated H2Bv3 knock-ins as they had a major growth defect, as suspected by the reviewer. Both the nuclear localization of the green fluorescence and growth defect suggest that H2Bv3 was correctly targeted.

(2) We also performed an additional experiment to specifically address the risk of cross-integrations when generating double genome edits (Lines 693-696, Figure S16). We used *D. discoideum* AX2 *act5::mCherry-P2A-act5*, *ecmA::mNeonGreen-P2A-ecmA* to create a double knock-out experiment targeting *mCherry* and *mNeonGreen* using donor oligos with FLAG- and Myc-tags respectively. By sorting directly for *mCherry* knock-outs and analysing those with both PCR and Sanger sequencing, we confirmed that all knock-outs had the

expected FLAG-tag (except for one knock-out that was caused by NHEJ). We did not observe a single cross-integration of a Myc-tag, suggesting that cross-integrations are rare, as long as homology arms are sufficiently distinct.

2.6. L682-686: Previous research by Muramoto and colleagues demonstrated the successful simultaneous editing of up to five genes, while they developed a convenient expression vector that enables dual-site editing. If the authors wish to highlight the efficiency or scalability of their method for multiplexed editing, it would be beneficial to compare their findings with these earlier studies. Furthermore, since the present approach involves the co-delivery of multiple donor oligonucleotides, the potential for interference among them should be considered, particularly when targeting more than two loci.

We now re-did the double knock-in experiment targeted *ecmA* and *act5* to confirm the efficiency of our approach. When sorting for *act5:mCherry-P2A-act5* knock-ins, we obtain a 59% knock-in efficiency of *mNeonGreen* at the *ecmA* locus (Figure S14 and S15). We confirmed the integrations through both PCR validations and Sanger sequencing (File S7). In addition, we performed a double knock-out experiment (see also Comment 2.5), where we did not find any cross-integrations, suggesting that the risk of cross-integrations is rare as long as homology arms are sufficiently distinct.

We think our RNP-based approach complements the plasmid-based protocol of Muramoto and colleagues (see also Comment 2.2), by allowing for complex edits in non-axenic strains and avoiding the need for cloning and antibiotic selection, thereby saving considerable time. However, when aiming to edit more than two loci, our approach is particularly powerful in combination with sorting (Lines 696-698), whereas for the plasmid-based approach antibiotic selection would be sufficient. Thus, depending on the use case, either one of these two approaches might be preferred. For instance, when building multiple expression reporters, our approach could save considerable time, while for combinatorial knock-outs the plasmid-based method could be superior.

2.7. While the advantages of the method are well described in the Discussion, its limitations are not sufficiently addressed. In particular, although the addition of donor oligonucleotides clearly increases knock-out efficiency, it may also decrease the likelihood of isolating in-frame or frameshift mutations in essential genes, as noted by You et al. 2020. For example, in donor-free genome editing targeting essential genes such as "tor", knock-out mutants could not be obtained; however, in-frame and frameshift variants were successfully isolated and analysed (Li et al., 2016; Ogasawara et al., 2022). Therefore, it would be beneficial for the authors to consider and briefly discuss the potential limitations of combining RNP with donor oligos, particularly in the context of editing essential genes.

Thank you for raising this point. Based on our validation experiments, mis-integrations seem to be rare. However, when targeting essential genes these are indeed expected, simply because correct integrations are lethal when generating knock-outs. We do not think this is a particular problem of the RNP-based method, but rather connects to the downstream fitness effects of genome edits, which have to be taken into account when developing editing strategies whether or not a repair template is involved. As long as one is cautious and performs the necessary validation experiments (PCR and Sanger sequencing), downstream

fitness effects can easily be identified. We therefore now highlight the need for such validation experiments in the discussion (Lines 792-794).

Minor points

2.8. Introduction: The authors should carefully distinguish between traditional gene manipulation via homologous recombination and CRISPR-based genome editing. For example, line 49 refers to conventional methods as "genome editing," while lines 66-67 describe CRISPR specifically. Since Schaap and colleagues used traditional recombination rather than CRISPR, careful use of terminology would help avoid potential confusion. Moreover, it would be helpful for the Introduction to more clearly acknowledge that previous comparative analyses were conducted using strains generated by traditional knock-out methods. Highlighting known limitations in *P. pallidum*, such as the low efficiency of gene disruption and the restricted range of available selection markers, would further underscore the value and advantages of the approach presented in this study.

We now revised the text to highlight when we refer to genome editing through classical homologous recombination versus HDR-mediated repair with our CRISPR approach (e.g., Lines 57, 420 and 743). We thank the reviewer for also pointing out the important research on *P. pallidum* and now acknowledge this work in our introduction by citing several of the corresponding papers. We decided not to single out *P. pallidum* in the introduction, because we think that most genome editing challenges are common among *Dictyostelid* species and therefore prefer to focus on the broader challenges.

2.9. The Methods section is somewhat redundant, as detailed protocols are also provided in the Supplementary file. It may improve readability to restructure this section by retaining only the most essential information in the main text.

We now reduced redundancies between the SI and method to improve readability.

2.10. The use of cell sorting for clonal isolation is a valuable aspect of this study, particularly effective when fluorescent proteins are expressed. However, in the case of standard gene knock-outs without a fluorescent marker, sorting serves primarily as a general cloning tool. To address a broader range of applications, it would be helpful for the authors to also consider and compare traditional plaque-based cloning on bacterial lawns.

We have not performed a direct comparison with classical plaque-based assays, but we now included a *pkaC* knock-out experiment (Figure S12) – which was previously targeted using plasmid-based CRISPR editing in combination with plaque-based assays – and we could indeed easily isolate knock-outs by screening for wells with arrested fruiting body development (Figure 5A). All these mutants had the expected knock-out mutations, based on PCR and Sanger sequencing. In the case of *pkaC*, both isolation approaches are therefore effective. When isolating knock-outs with strong growth defects compared to wildtype cells, or when generating fluorescent knock-ins, sorting would be a superior isolation method.

2.11. L190: The *act5* target, it appears that no such spacer without potential off-target sites could be identified. It would therefore be more accurate to revise this statement to reflect the actual design constraints.

Thank you! We now adjusted our statement.

2.12. L210: (4)  (5)

Corrected

2.13. L212: The manuscript describes two approaches for preparing the RNP complex, with crRNA and tracrRNA either added separately at a final concentration of 22 μM , or pre annealed and used at 15 μM . However, the reason for this difference in concentration is not clearly explained. Additionally, it is unclear which of these two methods was used in each experiment.

When we started developing our protocol, we did not include the pre-annealing step, but after approximately two years we discovered that this step further improves editing efficiencies (Figure S2). We therefore recommend including this step (Text 2). We now mention in each figure caption when a pre-annealing step was not included to make clear when we applied sub-optimal editing conditions (see also Lines 198-209). For the pre-annealing step, we followed the recommended protocol of IDT, which uses a somewhat lower concentration of 15 μM crRNA and tracrRNA. Please note that this concentration is still well above the 12 μM SpyCas9 concentration. The pre-annealing step is therefore expected to improve the folding of the gRNA complex but is not expected to affect the loading of SpyCas9 otherwise (in all cases, the gRNA concentration is higher than the SpyCas9 concentration when assembling the RNP complex).

2.14. L305: The species names are conventional names, whereas the rest part of the manuscript uses the nomenclatures proposed in ref. 5.

Corrected

2.15. L423-426: The use of the RNP complex in Dictyostelium is a key factor enabling a selection-free genome editing approach. While this point is mentioned in parts of the manuscript, it would benefit from being stated more clearly and emphasised, as it represents a significant advantage of the method.

Good point. We now added this information repeatedly throughout the manuscript (e.g., Lines 503, 548, and 783).

2.16. In Figure S2, knock-out appears to be achieved using a donor oligo, whereas lines 433-437 and Figure S3A-B seem to describe an NHEJ-based approach. As this could lead to confusion, it should be clarified which strategy was used in each case.

We now moved our reference to Figure S2 (now Figure S4) to a more appropriate sentence in the paragraph.

2.17. L434: It is unclear how many clones were analysed in total, and whether the results shown in Figure S3 represent all analysed clones or only selected successful ones. If the labels such as "MC1_" represent clone names, the numbering in Figure S3C seems inconsistent, as "3" appears twice.

Thank you for pointing out the error in Figure S3C, we now corrected this. For the validation experiments in Figure S3, we picked a small number of random clones and showed the corresponding Sanger sequencing results. For experiments with donor oligo, we purposely showed HDR-mediated knock-outs, and not NHEJ-mediated knock-outs, which occasionally occur as well. In addition to Figure S3, we now added >100 additional Sanger sequencing results in File S7.

2.18. L460: The data indicate that 56 bp arms actually yielded higher knock-out efficiency. It would be advisable to revise the wording to reflect this more precisely.

Given the error margins, we cannot conclude from Figure 1 that editing efficiencies for 56bp and 28bp homology arms are significantly different ($75\pm 4.6\%$ and $79\pm 1.4\%$). As such, we think it is accurate to say that editing efficiencies already peaked for 28bp homology arms.

2.19. L486-487: The term "PCR template" should be corrected to "PCR product".

Corrected.

2.20. L528: Schaap and colleagues previously incubated non-axenic strains in HL5 medium for one day prior to transformation to reduce nuclease activity, which may be elevated during bacterial growth. Since this step appears to have been omitted in the present study, it would be beneficial to discuss whether this omission could have affected transformation efficiency or genome editing outcomes.

Great suggestion! We now performed the suggested experiment and indeed observed improved knock-in efficiencies for *D. discoideum* NC4 when including a pre-culturing step on HL5 (Figure 3E; Lines 543-549). Alongside this observation, we also noticed a reduced collateral knock-out rate, which sparked a more in-depth analysis on these knock-outs. We redid our knock-in experiments and sequenced a large number of collateral knock-out mutants (Figure S9, File S7). From the sanger sequencing data, it became evident that collateral knock-outs were caused by either NHEJ, as observed before, or partial integration of the donor PCR product. Partial integrations were biased towards the 5' end of the donor PCR product (Figure S9E). We see two possible causes for this partial integration. It could be that nuclease activity causes degradation of the donor PCR product and thereby lowers full-length integrations. In addition, given that short homology arms are sufficient for HDR (Figure 1), it could also be that partial integrations are mediated by small sequence similarities between the CDS of the *mCherry* target site and *mNeonGreen* donor PCR product. We now highlight our new findings in the results (Lines 489-498). While exploring the collateral knock-out mutants, we also noticed a small mistake in how we calculated the knock-out rates for Figure 2 and S9 (Previously Figure S8), which is now correct. We apologize for this mistake.

2.21. L528, L531: The sentences describe transfections done with bacterially grown cells, but not transfections done in the presence of bacteria.

We now largely revised this section to clarify our experimental setup.

2.22. Figure 4, L548-576: It may seem unusual for a control experiment related to sorting to be presented as a main figure. The novelty and significance of this figure could be more explicitly stated.

We now revised the text to better highlight the novelty of our sorting protocol (Lines 576-579). In addition, we included a *pkaC* knock-out experiment (Figure S12) to illustrate how our sorting protocol can be used in the absence of fluorescent reporters.

2.23. Fig5D: In the *act5::mNeonGreen* reporter, fluorescence appears diminished at the anterior region of the slug. It would be beneficial to clarify whether this apparent non-uniformity is a result of projection artifacts or if it reflects a known characteristic of mNeonGreen, such as the uneven fluorescence observed during late developmental stages (Hashimura et al., 2024).

As we see the same for the *act5* reporter, at both the anterior and posterior region, we think this diminished intensity is largely an artifact of how we perform the maximum projections across the slug's axis. At the anterior and posterior there are fewer cells, so the maximum projection is expected to yield lower numbers.

2.24. L708: ref 49 & 63 describe the generation of knock-out cells via homologous recombination, rather than knock-in approaches.

Since these knock-outs were generated by knocking in an antibiotic resistance cassette at a specific target site, we still consider them bona fide knock-ins. However, we have revised the text (Lines 741-744) to clarify that homologous recombination was used in these cases, as opposed to CRISPR-based methods.

2.25. L710: The reported knock-in efficiency in *D. discoideum* (16%) is somewhat unclear, as it is not specified which experiment or figure this value is derived from.

This percentage was obtained by knocking in *mNeonGreen-P2A* in the *act5* locus of *D. discoideum* NC4 using non-axenic growth conditions (*E. coli* B/r suspension in KK2). Knock-in efficiencies for NC4 are somewhat lower than for AX2. We now explicitly state that this percentage belongs to *D. discoideum* NC4 strain (Lines 746).

2.26. L712: ref35 reports optimisation of the electroporation buffer specifically for cells cultured with bacteria. In the present study, the H50 buffer was used, but it would be helpful to include a brief discussion of this choice and whether it may have influenced transfection efficiency.

That is an excellent point. We have not systematically tried different buffers, but we agree that these might have an effect as well. Given that we have not tested different buffers, we prefer not to speculate about their potential impact in the discussion, but have added a sentence in the discussion encouraging researchers to try using electroporation buffers and conditions that have worked for their strains of interest (Lines 845-849).

2.27. Figure 7: Given that *D. discoideum* possesses over 20 actin genes, and genome analysis indicates that other *Dictyostelids* also have multiple actin isoforms, it is essential to

verify that integration has occurred at the intended locus. Referring to the outcome as a "knock-in" without such confirmation risks being misleading.

We have confirmed that knock-ins occurred at the expected locus using Sanger sequencing. This data is now included in File S7.

2.28. L749: The statement that cell stocks can be prepared "a few days later" may somewhat overstate the general applicability of the approach. While this timeline may be feasible for fluorescence-based control experiments, confirming knock-out or knock-in clones through genome sequencing would typically require more time.

We appreciate that this is a relatively optimistic timeline and now adjusted the sentence.

2.29. Figure S10C: The sum of the displayed percentages is 101%.

Thank you for pointing out this rounding error. Corrected.

Reviewer 3

In this exciting study, the authors developed a method to rapidly generate CRISPR mutants in axenic, non-axenic, and phylogenetically distant *Dictyostelid* species. Their approach was to deliver the CRISPR/Cas 9 ribonucleotide complex (RNP) into the amoeba via electroporation instead of using all-in-one CRISPR plasmids. The RNP delivery circumvented the need for cloning unlike with the CRISPR vector approach, shortening the time to generate mutants. Additionally, the authors used single cell sorting ~2-3 days after transfection to bypass time-intensive selection and screening of polyclonal populations and generating monoclonal populations by limiting dilutions.

As proof of concept, the authors optimized the workflow and parameters in axenically-grown AX2 cells. Then, they performed genome editing with amoeba in non-axenic growth conditions. They made clever use of fluorescent strains to perform knock-outs and knock-ins with repair templates and used flow cytometry to provide quantitative data for genomic editing efficiencies. Next, the authors edited multiple genes simultaneously under axenic conditions. Finally, the authors generated knock-outs of 6 diverged *Dictyostelids*, including 3 previously intractable species. By eliminating the cloning and selection steps, as well as the reliance on limited selectable markers, this study dramatically increases the genetic tractability of this model organism, as well as several other *Dictyostelium* species.

General remarks

There are currently a very limited set of tools for genomic editing of non-axenic *Dictyostelium*. This manuscript identifies clear gaps in this field and tackles them directly through technical innovation. The authors were extremely thorough, covering knock ins, knock outs, fluorescent reporters, endogenous fusion proteins, rates of NHEJ vs HR, effective ways to isolate clones, and combinations of mutations. In line with their thoroughness, the authors included the nanopore sequencing and assembly of the AX2 act5::mCherry strain to look for off-target edits. This is above and beyond where the field has been to this point. This work would expand the toolset for genomic editing of *Dictyostelium* species, including non-axenic strains, and has the potential to become a new guidebook for

researchers in the field. Additionally, the detailed information in the methods and in the supplement is very well written and will help other labs use this method successfully. This manuscript will interest most researchers working with *Dictyostelium* as these methods have downstream applications to many if not all facets of *Dictyostelium* biology. I believe that the data shown supports most of the conclusions stated in the text. A few changes to the text and data presented could strengthen the manuscript even further (suggested below).

We thank the reviewer for their supportive feedback and did our best to address the comments below.

Major points

3.1. The authors were able to get purified SpyCas9 from their own institution's facility. To enable broader uptake of this method, the authors do describe a method for purification, but many *Dictyostelium* labs do not have this set up available. It is also disappointing that many of the commercial SpyCas9s perform so poorly in Fig S5.

We were also surprised to see the stark difference between commercial SpyCas9s and our in-house ultrapure SpyCas9. We have insufficient information from the commercial providers to exactly pinpoint what explains the difference, but from our analysis, it is clear that both the purity and the 'version' of SpyCas9 are critical. Fortunately, the TrueCut Cas9 Protein v2 (Thermo Fisher A36498) still shows relatively high efficiencies and can therefore be used without any further adjustments to our protocol. We now demonstrate this more extensively by including additional data on knock-outs (Figure S7), single- (Figure S2, S14, S15) and double knock-ins (Figure S14, S15). We are also encouraged by the fact that we were already contacted by researchers in the field that have successfully applied our RNP-based protocol (based on our preprint) using commercial SpyCas9 to generate knock-ins.

3.1a. Do the authors have any suggestions (or data) about optimizing these protocols with commercial SpyCas9 sources?

We have not systematically tried to improve the performance of commercial SpyCas9s, but a new titration experiment (Figure S7) suggests that increasing the SpyCas9 concentration could improve efficiencies, although this might come with the risk of off-target edits. In addition, we think it might be worth exploring different buffers for both the RNP mix and transfection mix (see also Comment 2.26).

3.1b. Additionally, as this is likely the most expensive part of the protocol for most researchers, have the authors titrated how much SpyCas9 is needed in these reactions?

We have now performed a titration experiment with both our in-house ultrapure SpyCas9 and the TrueCut SpyCas9 Protein v2 (Thermo Fisher A36498) (Figure S7). These results show that (1) lower SpyCas9 concentrations reduce the knock-out efficiency and (2) that we do not saturate the editing efficiencies at the SpyCas9 concentration that we currently use in our protocol. In other words, editing efficiencies can likely be improved further by increasing the SpyCas9 concentration, if one controls for unwanted off-target nuclease activity.

3.1c. Is it possible to successfully perform more complex genomic editing (as shown in Fig 5) with commercial Cas9 from Thermo? Although the edit shown are impressive, a more

thorough investigation of the commercial proteins would help other researchers to know which of the techniques presented here would plausibly work in another lab.

We have now redone both the single (Figure S2) and double knock-in (Figure S14, S15) experiments using both our in-house ultrapure SpyCas9 and TrueCut SpyCas9 Protein v2. In all cases we find two to three times lower editing efficiencies for the commercial SpyCas9, but that did not affect our ability to obtain the desired mutants. In fact, even for the double knock-in mutant targeting *act5* and *ecmA*, we got a 20% efficiency with the commercial SpyCas9 and could easily isolate the desired mutants (Figure S15). Our RNP-based protocol could therefore easily be applied with commercial SpyCas9 for generating complex genome editing.

3.2. The authors conclude that their system can be used to successfully generate double knock-ins and simultaneous knock-in and knock-out mutations. I agree that the single knock-out rates are very high, but I encourage the authors to moderate their language for the other methods, as the rates are much lower. Additionally, the following edits would provide clarification and further support for the authors conclusions:

There are three causes why we sometimes report lower efficiencies: (1) there is always batch-to-batch variation across experiments (see Figure S1 and S6); (2) some CRISPR-Cas9 targets consistently provide lower editing efficiencies, even under optimal conditions (Figure 6A, S6 and S12), (3) for some experiments we did not use the optimal protocol. For instance, in a fraction of the experiments we did not apply the pre-annealing step of the gRNA (see Methods), since we only discovered relatively late in developing our protocol that this further improves efficiencies (see Figure S2 for comparison). We now explicitly mention in each figure caption when we deviate from the optimal protocol.

Besides updating the figure captions, we also explicitly state in the abstract and discussion that we refer to maximum efficiencies when mentioning the 90% knock-out and 50% knock-in efficiencies. Having said that, even for complex constructs, the RNP-based protocol can support remarkably high editing efficiencies. For instance, in the newly performed double knock-in experiments (performed under optimal conditions) targeting *act5* and *ecmA*, we obtained 60% knock-in efficiency at the *ecmA* locus when sorting for *mCherry* knock-ins at the *act5* locus (Figure S14 and S15). Thus, without antibiotic selection, the majority of the isolates have the expected double knock-in mutation in *D. discoideum*. For other *Dictyostelid* species we never obtained such high efficiencies, but we aim to improve this in the future.

3.2a. The methods used in Figure S10 are currently unclear. A targeting and outcome schematic similar to Figure 2A would be very helpful for clarification.

We now added schematics to all these figures (Figure S14, S16 and S17).

3.2b. Because *ecmA* is not expressed in vegetative cells, we cannot assess its editing in S10A and S10C. To appreciate the changes in fluorescence resulting from genomic editing, it would help to see flow cytometry data (or similar) from disaggregated slugs for similar conditions as S10A-C. Alternatively, sequencing data could be used to assess the editing frequency in the *act5* and *ecmA* loci.

We thank the reviewer for this feedback. We now redid the double knock-in experiment (Figure S14 and S15) and sequenced all mCherry+ sorted cells. This revealed a 59% *mNeonGreen* knock-in efficiency at the *ecmA* locus. We also performed sequencing on controls cells that were sorted blindly, showing an *mNeonGreen* knock-in efficiency of 43%.

3.2c. Despite concluding that successful mCherry knock-out occurred, panels S10D-F show that mCherry cells are still very bright (at 10^4 after knock-out), this is at odds with the "no fluorescent" labelling in the figures. Please explain. Is it possible that the labelled actin has not turned over at the time of data collection?

Thank you for pointing out this confusion. The flow cytometry data for the double knock-in is acquired with the BD FACSAria Fusion Flow Cytometer (for sorting) and that of the combined knock-in and knock-out with the BD FACSymphony A3 analyzer. The absolute fluorescent intensities are therefore not directly comparable between experiments. As we always perform our flow cytometry experiments with positive and negative controls, we are confident about our non-fluorescent labelling. Since we redid some of these experiments, the figures are now partly rearranged, but for transparency we now mention in the associated figure captions what flow cytometer was used for data acquisition.

3.3. The authors conclude that they generated mutants of 6 diverged *Dictyostelids* without major impacts to growth and development. However, not much raw data was shown for this section of the manuscript, unlike all other sections. To strengthen the conclusions in this section, I recommend:

Please see comments below.

3.3a. including the flow cytometry data for the strains and the images of the plates after sorting (or cell count data to assess growth).

We now added the raw flow cytometry files to the manuscript (File S8) as well as the Sanger sequencing results (File S7), which confirm the expected knock-in mutants for each of the edited *Dictyostelid* species. For each species, we furthermore have detailed time lapse microscopy (File S5 and File S6).

3.3b. including data such as images of non/mock electroporated cells at the same timepoint as the images shown in 5B to illustrate whether development or growth has been affected.

This is an excellent point. For each species, we performed a side-by-side comparison between strains with and without fluorescent reporters using time lapse microscopy. Time lapses were examined at low and high magnifications, which are included in Files S5 and S6. Importantly, we did not detect any noticeable phenotypic defect in the presence of the fluorescent reporter.

3.3c. explaining why false color was used as a measure of fluorescence in panel 5B and not in panels 4C-D and 5A-B.

Since slugs and fruiting bodies have a much higher fluorescent intensity than the feeding front, we falsely coloured the green fluorescence using an inferno colour scale to accurately visualize the entire dynamical range of fluorescent intensities.

Minor points

3.4. Line 52- the author's name was misspelled, it is Muramoto.

Thank you for pointing out this typo. Corrected

3.5. Line 55- Muramoto's system can be used with 2 guides depending on the vector.

Corrected.

3.6. Line 69-Paschke et al should be cited here as this is one existing sample of a CRISPR system for non-axenic Dicty.

We cite the work of Paschke paper elsewhere in the manuscript, but in this particular line, we think it would be somewhat confusing as – to our knowledge – Paschke did not use CRISPR in her protocol (<https://doi.org/10.1371/journal.pone.0196809>).

3.7. Line 196-what is the FAB additive to the HL5? What is its concentration?

We added a reference to Text S2, which details all buffers and media.

3.8. Methods text has life and dead cells as DRAQ7 negative. I believe only live cells would be negative for this stain.

Thank you for pointing out this mistake. This is now corrected.

3.9. For panel 1D-use open or different symbols to clarify/differentiate this panel from the others.

Great idea, we implemented this suggestion. When adjusting the figure, we noticed a mistake in Figure 1C, where the data points for the 14bp homology arms were shifted. We apologize for this mistake and now corrected this as well.

3.10. The X axis labelling for 6A is unclear, explicitly stating N or C tag would be more clear.

Thanks for this suggestion, we now added this information

3.11. Line 504-no citations or statistics to demonstrate that this is the highest ever recorded genetic editing without selective markers. Please add citations and include what the recorded efficiencies were in the other papers, this could be in a table format. Relatedly, knock-in efficiencies of ~15-20% are labelled as high, please expand and discuss why this is with relevant comparisons and citations.

Thank you for pointing this out. A direct comparison with previous protocols is indeed somewhat problematic, since our protocol is selection-free and efficiencies relate to both

transfection and editing efficiencies, while in previous protocols there is also the efficiency of antibiotic selection (see also Comment 2.2). Given this complication, we reformulated our statement. Because our method does not rely on an antibiotic marker, our reported knock-out and knock-in efficiencies are remarkably high.

3.12. Panel 5E does not show the images of the wells although they were referenced in the text. This can be addressed by including the images of the wells or rewriting of the sentence in the results to indicate that 5E shows one representative image of a slug.

We have now adjusted the text (Lines 684-696). We have also redone the double knock-in experiment (Figure S14 and S15) and included more of the image data.

20th Oct 2025

Manuscript Number: MSB-2025-13091R
Title: Unlocking CRISPR-Cas9 editing for widely diverse Dictyostelid species
Author: Mireia Garriga-Canut
Nikki Cannon
Matt Benton
Andrea Zanon
Samual Horsfield
Jacob Scheurich
Kim Remans
John Lees
Alexandre Paix
Jordi van Gestel

Dear Dr. van Gestel,

Thank you again for submitting your work to Molecular Systems Biology. We have now heard back from the three original referees. As you will see, the referees find your work exciting and strongly support publication. Yet, our editorial assistance team flagged a few remaining formatting and technical issues that need to be corrected before final acceptance. I provide their comments below.

Please resubmit your revised manuscript online within one month and ideally as soon as possible. Please use the Manuscript Number (above) in all correspondence.

When you resubmit your manuscript, please download our CHECKLIST (<https://bit.ly/EMBOPressAuthorChecklist>) and include the completed form in your submission. *Please note* that the Author Checklist will be published alongside the paper as part of the transparent process (<https://www.embopress.org/page/journal/17444292/authorguide#transparentprocess>)

Click on the link below to submit your revised paper.

Yours sincerely,

Yehu Moran
Academic Editor
Molecular Systems Biology

If you do choose to resubmit, please click on the link below to submit the revision online before 19th Nov 2025.

IMPORTANT: When you send your revision, we will require the following items:

1. the manuscript text in LaTeX, RTF or MS Word format
2. a letter with a detailed description of the changes made in response to the referees. Please specify clearly the exact places in the text (pages and paragraphs) where each change has been made in response to each specific comment given
3. three to four 'bullet points' highlighting the main findings of your study
4. a short 'blurb' text summarizing in two sentences the study (max. 250 characters)

5. a 'thumbnail image' (550px width and max 400px height, Illustrator, PowerPoint or jpeg format), which can be used as 'visual title' for the synopsis section of your paper.

6. Please include an author contributions statement after the Acknowledgements section (see <https://www.embopress.org/page/journal/17444292/authorguide#manuscriptpreparation>)

7. Please complete the CHECKLIST available at (<https://bit.ly/EMBOPressAuthorChecklist>). Please note that the Author Checklist will be published alongside the paper as part of the transparent process (<https://www.embopress.org/page/journal/17444292/authorguide#transparentprocess>).

See also figure legend guidelines: <https://www.embopress.org/page/journal/17444292/authorguide#figureformat>

9. Please note that corresponding authors are required to supply an ORCID ID for their name upon submission of a revised manuscript (EMBO Press signed a joint statement to encourage ORCID adoption).

(<https://www.embopress.org/page/journal/17444292/authorguide#editorialprocess>)

Currently, our records indicate that the ORCID for your account is 0000-0001-5598-5239.

Link Not Available

10. Include a Reagents and Tools Table as part of the Methods section, which can be downloaded from our author guidelines (<https://www.embopress.org/page/journal/17444292/authorguide#structuredmethods>)

*** PLEASE NOTE *** As part of the EMBO Press transparent editorial process initiative (see our Editorial at <https://dx.doi.org/10.1038/msb.2010.72> , Molecular Systems Biology will publish online a Review Process File to accompany accepted manuscripts. When preparing your letter of response, please be aware that in the event of acceptance, your cover letter/point-by-point document will be included as part of this File, which will be available to the scientific community. More information about this initiative is available in our Instructions to Authors. If you have any questions about this initiative, please contact the editorial office (msb@embo.org).

comments from editorial assistance team

*MANUSCRIPT FORMAT (.docx; figures; section order): please make sure to upload a final .docx file, no figures, no track changes.

*FUNDING: missing info in our system: EMBL and iXcore - iXlife - iXblue foundation

*Keywords (up to 5): missing, please provide

*REFERENCE FORMAT: incorrect, should be listed alphabetically

*Author Contributions/CRediT: section needs to be removed from the manuscript text and submitted only via the submission system.

*FIGURE CALLOUTS: all callouts should be listed sequentially; missing callouts for Fig. 3E, 4A, 5B-C and 7A. Please correct.

*APPENDIX 1 FILE WITH ToC: Appendix file needs to be in PDF format; title page should contain "Appendix for + ms title" and ToC with the page numbers for the listed items; nomenclature should be Appendix Figure Sx and Appendix Table Sx throughout ms and Appendix PDF (please do not use the term "Supplementary")

*SYNOPSIS IMAGE: missing, please provide according to instructions.

*SYNOPSIS TEXT: missing, please provide according to instructions.

Reuse between Figure 5C and Figure S13B, second panel. (Figure 5 is referenced in the figure legend, but it is not explicitly called out). Please clarify in the text and consult with us in case of doubt.

*DATA CHECK: FAILED - Please note that reviewer access code for Zenodo dataset is not provided in the data availability statement. In any case towards publication it is necessary the authors will make sure all data becomes publicly available and accessible.

Extra Notes:

- "Materials and Methods" should be renamed to "Methods"
- Sections need to be named and the order should be corrected: Title page - Abstract - Keywords - Introduction - Results - Discussion - Methods - Data Availability - Acknowledgements - Disclosure and Competing Interests Statement - References - Figure Legends - Table(s) - Expanded View Figure Legends.

Reviewer #1:

The authors have addressed all my concerns

Reviewer #2:

The authors have adequately addressed the reviewer's comments through additional experiments, clarifications, and revisions to the manuscript. I have no further concerns.

Reviewer #3:

The authors have provided very useful responses to all of my comments. In fact, I feel they have gone above and beyond many of the requests made by myself and the other reviewers. They are to be commended on an excellent manuscript!

The authors addressed the editorial issues.

8th Nov 2025

Manuscript Number: MSB-2025-13091RR

Title: Unlocking CRISPR-Cas9 editing for widely diverse Dictyostelid species

Author: Mireia Garriga-Canut

Nikki Cannon

Matt Benton

Andrea Zanon

Samual Horsfield

Jacob Scheurich

Kim Remans

John Lees

Alexandre Paix

Jordi van Gestel

Dear Dr. van Gestel,

First, I sincerely apologize for the delay with your paper. It is due to a technical error and completely on me as an editor. Thank you again for submitting your work to Molecular Systems Biology. There are a couple of minor technical issues that require your attention before your paper can be formally accepted for publication:

1. The synopsis text is missing bullet points. Please correct this. You can look on examples for recently published manuscripts on our website.
- The synopsis image you provided is too small and should be resized to exactly 550 pixels wide and 300-600 pixels high.

Please resubmit your revised manuscript at your earliest convenience so we can proceed with the acceptance of your manuscript.

Click on the link below to submit your revised paper.

Yours sincerely,

Yehu Moran

Academic Editor

Molecular Systems Biology

If you do choose to resubmit, please click on the link below to submit the revision online before 8th Dec 2025.

IMPORTANT: When you send your revision, we will require the following items:

1. the manuscript text in LaTeX, RTF or MS Word format
2. a letter with a detailed description of the changes made in response to the referees. Please specify clearly the exact places in the text (pages and paragraphs) where each change has been made in response to each specific comment given
3. three to four 'bullet points' highlighting the main findings of your study
4. a short 'blurb' text summarizing in two sentences the study (max. 250 characters)
5. a 'thumbnail image' (550px width and max 400px height, Illustrator, PowerPoint or jpeg format), which can be used as 'visual

title' for the synopsis section of your paper.

6. Please include an author contributions statement after the Acknowledgements section (see <https://www.embopress.org/page/journal/17444292/authorguide#manuscriptpreparation>)

7. Please complete the CHECKLIST available at (<https://bit.ly/EMBOPressAuthorChecklist>). Please note that the Author Checklist will be published alongside the paper as part of the transparent process (<https://www.embopress.org/page/journal/17444292/authorguide#transparentprocess>).

See also figure legend guidelines: <https://www.embopress.org/page/journal/17444292/authorguide#figureformat>

9. Please note that corresponding authors are required to supply an ORCID ID for their name upon submission of a revised manuscript (EMBO Press signed a joint statement to encourage ORCID adoption).

(<https://www.embopress.org/page/journal/17444292/authorguide#editorialprocess>)

Currently, our records indicate that the ORCID for your account is 0000-0001-5598-5239.

Link Not Available

10. Include a Reagents and Tools Table as part of the Methods section, which can be downloaded from our author guidelines (<https://www.embopress.org/page/journal/17444292/authorguide#structuredmethods>)

*** PLEASE NOTE *** As part of the EMBO Press transparent editorial process initiative (see our Editorial at <https://dx.doi.org/10.1038/msb.2010.72> , Molecular Systems Biology will publish online a Review Process File to accompany accepted manuscripts. When preparing your letter of response, please be aware that in the event of acceptance, your cover letter/point-by-point document will be included as part of this File, which will be available to the scientific community. More information about this initiative is available in our Instructions to Authors. If you have any questions about this initiative, please contact the editorial office (msb@embo.org).

All editorial and formatting issues were resolved by the authors.

24th Nov 2025

Manuscript number: MSB-2025-13091RRR

Title: Unlocking CRISPR-Cas9 editing for widely diverse Dictyostelid species

Dear Dr. van Gestel,

Thank you again for sending us your revised manuscript. We are now satisfied with the modifications made and I am pleased to inform you that your paper has been formally accepted for publication.

Yours sincerely,

Sincerely,

Yehu Moran

Editor

Molecular Systems Biology
